# Entropic Desired Dynamics for Intrinsic Control

**Steven Hansen**[*]  **Guillaume Desjardins**  **Kate Baumli**  **David Warde-Farley**
DeepMind          DeepMind          DeepMind          DeepMind

**Nicolas Heess**          **Simon Osindero**          **Volodymyr Mnih**
DeepMind                  DeepMind                  DeepMind

## Abstract

An agent might be said, informally, to have mastery of its environment when it has maximised the effective number of states it can reliably reach. In practice, this often means maximizing the number of latent codes that can be discriminated from future states under some short time horizon (e.g. [15]). By situating these latent codes in a globally consistent coordinate system, we show that agents can reliably reach more states in the long term while still optimizing a local objective. A simple instantiation of this idea, **E**ntropic **D**esired **D**ynamics for **I**ntrinsic **C**onTrol (EDDICT), assumes fixed additive latent dynamics, which results in tractable learning and an interpretable latent space. Compared to prior methods, EDDICT's globally consistent codes allow it to be far more exploratory, as demonstrated by improved state coverage and increased unsupervised performance on hard exploration games such as Montezuma's Revenge.

## 1   Introduction

Endowing reinforcement learning agents with the ability to learn effectively from unsupervised interaction with the environment, i.e. without access to an extrinsic reward signal, has the potential to make reinforcement learning practical in settings where the tasks the agent will face are initially unknown or where task feedback is expensive. The natural question is: what should the agent learn in the absence of extrinsic rewards? One appealing guiding principle is maximizing the number of states the agent can reach and to which it can reliably return.

Intrinsic control methods have shown promise in this direction. By maximizing the mutual information between a latent code $z$ and future states reached by a policy conditioned on this code, intrinsic control methods learn to map latent codes to behaviors from which the code can be inferred. One major limitation of such approaches is that the latent codes $z$ are usually sampled from a fixed prior distribution $p(z)$. Using a fixed prior means that such approaches are unable to learn codes that correspond to states that cannot be reached in the time horizon $T$, since any code can be sampled in any state. Simply increasing the time horizon $T$ does not solve the problem since it leads to a sparser learning signal. Learning a state-dependent prior has proven to be difficult and has been shown to lead to fewer learned codes/goal states [15]. This inability to learn how to reach distant states limits the usefulness of such intrinsic control approaches.

We propose to sidestep this limitation by replacing the fixed code distribution $p(z)$ with a fixed dynamics model over codes $p(z_t|z_{t-1})$. Our algorithm, **E**ntropic **D**esired **D**ynamics for **I**ntrinsic **C**onTrol (EDDICT), learns to map sequences of latent codes sampled from this dynamics model to behaviors for which the state transition dynamics in the environment match the latent code dynamics. EDDICT learns to map each $z_t$ to a state that is reachable from the state corresponding to $z_{t-1}$,

---

[*]Correspondence to stevenhansen@deepmind.com

35th Conference on Neural Information Processing Systems (NeurIPS 2021).

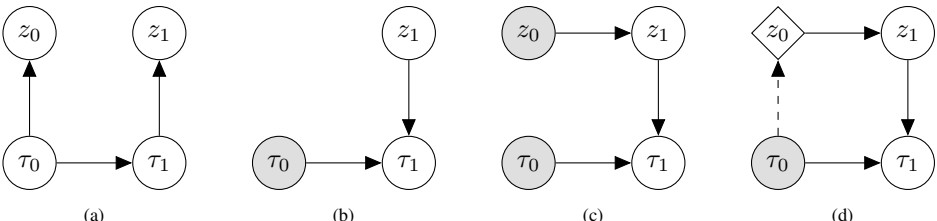

Figure 1: Graphical models for various priors and posteriors of interest. Circles denote random variables which are observed (shaded) or latent (white), with diamonds denoting deterministic quantities. (a) Prior over a particular trajectory consisting of two sub-trajectories $\{\tau_0, \tau_1\}$, and auxiliary variables $\{z_0, z_1\}$. (b) Posterior inference with independent codes, as in prior work. (c) Naive posterior inference for the sub-trajectory $\{z_1, \tau_1\}$, conditioned on the past. (d) Posterior inference with hindsight. Despite $z_0$ being observed, we infer $z_1$ based on the most likely code $z_0$ to have generated $\tau_0$, using the variational reverse predictor (dashed line).

allowing it to reach states much farther than the time horizon $T$ using sequences of codes $z$. We show that even highly constrained latent dynamics (i.e. additive noise) are sufficient to both interpret latent codes in terms of their corresponding locations in state space, and encourage exploratory behavior to a far greater extent when compared to prior methods.

## 2   Notation

Our environment is a special case of a Markov Decision Process (MDP) without rewards or terminal signals: $\mathcal{M} : (S, A, P, P_0)$. $S$ is the state space, $A$ is the action space, $P(s_{t+1} \mid s_t, a_t)$ the conditional distribution representing the state transition dynamics when taking action $a_t \in A$ from state $s_t \in S$ and $P_0(s)$ the initial state distribution. For simplicity, we present our method in the episodic setting with episodes of length $T = MK$, but relax this assumption in practice. Agents interact with the environment according to a policy $\pi_\theta(a \mid s)$ with parameters $\theta$, yielding trajectories $\tau = [s_0, a_0, s_1, a_1, \cdots s_T]$, and distributed as $p^{\pi_\theta}(\tau) = P_0(s_0) \prod_{t=0}^{T-1} P(s_{t+1} \mid s_t, a_t)\pi_\theta(a_t \mid s_t)$.

It will be useful for us to segment a given trajectory $\tau$ into sub-trajectories of length $K$, with $\tau = [s_0, \tau_0, \tau_1, \cdots]$, and $\tau_i = [a_{iK}, s_{iK+1}, a_{iK+1}, \cdots a_{(i+1)K-1}, s_{(i+1)K}]$. Note that $\tau_i$ is defined to include $s_{(i+1)K}$, but *not* the state $s_{iK}$ from which $a_{iK}$ was sampled. With a slight abuse of notation and denoting $\tau_{-1} := s_0$, we rewrite $p^{\pi_\theta}(\tau) = P_0(s_0) \prod_{i=0}^{M-1} p^\pi(\tau_i \mid \tau_{i-1})$, with $M$ the number of sub-trajectories per episode and $p^{\pi_\theta}(\tau_i \mid \tau_{i-1}) = \prod_{t=iK}^{(i+1)K-1} P(s_{t+1} \mid s_t, a_t)\pi_\theta(a_t \mid s_t)$. [2]

Hierarchical agents sample a high-level goal or latent variable $z \sim p(z)$ every $K$ steps, and interact with the environment via a parametric conditional low-level policy $\pi_\theta(a \mid s, z)$, which can be thought of as a fixed duration option [42]. Composing $p(z)$, $\pi_\theta(a \mid s, z)$ and transition dynamics yields an augmented trajectory $\Lambda = [s_0, z_0, \tau_0, z_1, \tau_1, \cdots]$ [3], whose distribution decomposes as $p^{\pi_\theta}(\Lambda) = P_0(s_0) \prod_{i=0}^{M} p(z_i)p^{\pi_\theta}(\tau_i \mid \tau_{i-1}, z_i)$, with $p^{\pi_\theta}(\tau_i \mid \tau_{i-1}, z_i)$ defined analogously to $p^{\pi_\theta}(\tau_i \mid \tau_{i-1})$ with conditional policy $\pi_\theta(a_t \mid s_t, z_i)$.

To simplify exposition, we index sequences at the timescale of sub-trajectories using index $i$, e.g. $[z_i, \tau_i, z_{i+1}, \tau_{i+1}, \cdots]$ and reserve index $t$ for indexing sequences at the granular timescale of actions, e.g. $[s_t, a_t, s_{t+1}, a_{t+1}, \cdots]$. Concretely, indexing by $i$ should be interpreted as $iK$, as in $s_i := s_{iK}$. For a general sequence $x = [x_0, x_1, \cdots]$, we define $x_{<t} := [x_0, \cdots, x_{t-1}]$ and extend this notation to augmented trajectories as follows: $\Lambda_{<i} := [s_0, z_0, \tau_0, \cdots, z_{i-1}, \tau_{i-1}]$.

## 3   Method

We would like to learn goal directed agents, which are capable of reaching any state $s \in S$, given a goal or state embedding $z \in Z$. Extending earlier work on empowerment [40, 34], Variational

---

[2]The dependence on $\tau_{i-1}$ is thus due to $s_{iK} \in \tau_{i-1}$.

[3]We avoid introducing new notation for the corresponding semi-MDP, as our present notation allows us to reason about sub-trajectories, for both standard policies $\pi(a \mid s)$ and conditional policies $\pi(a \mid s, z)$.

Intrinsic Control (VIC) [19] and related methods (e.g. [15]) propose to achieve this by learning a conditional policy $\pi(a \mid s, z)$ which maximizes $\mathcal{I}(z; \tau)$, the mutual information between the latent code $z$ and (possibly a subset of) the resulting trajectory obtained by following $\pi$. Unfortunately, this objective can be difficult to train in practice as we scale both the number of options and the horizon over which the code is executed [1, 15]. Our method addresses both of these issues in a principled manner by introducing temporal dependencies between a sequence of latent codes, evolving under simple linear dynamics, which decomposes the full objective into a sum of local mutual information lower-bounds, without loss of coherence of the global code.

Entropic Desired Dynamics for Instrinsic Control (EDDICT) can be understood from the perspective of divergence minimization [20, 13, 27]. Concretely, we can define a prior policy $\mu$ which induces a distribution $p^\mu(\Lambda)$ over the space of augmented trajectories. We then learn a (posterior) policy $\pi$ by minimizing the KL-divergence between $p^{\pi_\theta}(\Lambda)$ and $p^\mu(\Lambda)$.

## 3.1 VIC as Divergence Minimization

Given a prior policy $\mu$, we construct a prior distribution over an augmented space $(\tau, z)$, with auxiliary variables [3] $z \in Z$, as $p^\mu(z, \tau) = p^\mu(\tau) q_w(z \mid \tau)$. The conditional $q_w(z \mid \tau)$ is a learnt predictor, with parameters $w \in \Omega$, which aims to predict $z$ from the underlying trajectory.

We can show that an entropy regularized version of VIC is obtained by maximizing $\mathcal{O}_{\text{ent-VIC}}(\theta, w) = -\text{KL}\left[p(z) p^{\pi_\theta}(\tau \mid z) \| p^\mu(z, \tau)\right]$ wrt. the parameters of $\pi_\theta$ and $q_w$, with $p(z)$ a fixed or learnt distribution over options. Intuitively, we seek a code conditioned policy which generates trajectories having high probability under our trajectory prior, and from which $z$ can be inferred in hindsight. After some algebra, this simplifies to:

$$\mathcal{O}_{\text{ent-VIC}}(\theta, w) = \mathbb{E}_{\substack{z \sim p(z) \\ \tau \sim p^{\pi_\theta}(\tau \mid z)}} \left[ \underbrace{\log q_w(z \mid \tau) - \log p(z)}_{\text{ⓐ } \mathcal{I}_q(z; \tau)} - \sum_{t=0}^{T-1} \underbrace{\log \frac{\pi_\theta(a_t \mid s_t, z)}{\mu(a_t \mid s_t)}}_{\text{ⓑ regularizer}} \right]$$

In the above, $\mathcal{I}_q(z; \tau)$ refers to the variational lower-bound [2, 34] to the mutual information $\mathcal{I}(z; \tau) = \mathbb{E}[\log p^{\pi_\theta}(z \mid \tau) - \log p(z)]$, using reverse predictor $q$ trained to approximate the true posterior distribution $p^{\pi_\theta}(z \mid \tau)$. In expectation, the regularization terms correspond to a sum of KL-divergences between our conditional policy and the prior over actions.[4] The original objective $\mathcal{O}_{\text{VIC}}$ is obtained by dropping this regularizer and choosing the reverse predictor $q_w^{\text{VIC}} := q_w(z \mid s_0, s_T)$, which predicts $z$ from the first and last states of the trajectory.

Since we focus on discrete action spaces, we set $\mu$ to a uniform distribution over actions, causing ⓑ to revert to standard entropy rewards [48]. In practice, we optimize the above objective using a value-based reinforcement learning algorithm and $\epsilon$-greedy policies (in lieu of a Boltzmann policy), and thus omit these terms. Note that the auxiliary variable perspective of VIC can also be found in Hausman et al. [23].

## 3.2 Incorporating Temporal Dynamics

Instead of sampling a single goal to be reached within the duration of the episode, it may be preferable to sample a sequence of codes either as *relative (or local) goals*, parameterized relative to the agent's current position, or as *way points*, a sequence of global goal coordinates which the agent should visit in sequence.

**Relative vs Global Codes.** Local goals can be implemented for VIC by resampling a latent code every $K$ steps and maximizing a sum of local objectives of the type $\mathcal{I}_q(z_i; s_{i+1} \mid s_i)$, with option $z_i$ initiated from state $s_i$. We describe these codes as having *local semantics*, as an option $z_i$ should only be inferable in the context of the relationship between its initiation state $s_i$ and final state $s_{i+1}$. In essence, each $z_i$ represents a local displacement which the low-level policy should execute. In contrast, the strategy of sampling way points in some global frame of reference would require maximizing

---

[4]The regularization terms emerge from the fact that the transition dynamics are shared by both $p^\mu(\tau)$ and $p^\pi(\tau \mid z)$, and thus cancel out in the computation of the KL-divergence.

$\mathcal{I}_q(z_i; s_{i+1})$. Unfortunately, this would seem to require learning a state-dependent high-level policy which gives higher probability to goals $z_i$ which are reachable (in $K$ steps) from $s_i$.

Ours is a hybrid of these two approaches: by specifying goals relative to previously sampled codes, in the form of a Markov chain with simple linear dynamics, EDDICT can recover codes with global semantics while avoiding the need to explicitly train a high-level policy.

**EDDICT Prior.**  As in Section 3.1, we specify a joint distribution over the set of sub-trajectories $\{\tau_i\}$ and auxiliary variables $\{z_i\}$, $i \in [0, M-1]$. Recall that $\tau_{-1} := s_0$. Our prior for an augmented trajectory $\Lambda$ is given by:

$$p^\mu(\Lambda \mid s_0) = \prod_{i=0}^{M-1} p^\mu(\tau_i \mid \tau_{i-1}) q_w(z_i \mid \tau_i),$$

again with $\mu$ a uniform distribution over actions. As we shall see, making the *a priori* assertion that $z_i$ is conditionally independent of $\tau_{i-1}$ given $\tau_i$ will ensure that our objective breaks down as a sum of local objectives, amenable to greedy optimization. This prior is illustrated in Fig. 1a. Using the reverse predictor $q_w^{\text{EDDICT}} := q_w(z_i \mid s_{i+1})$, which predicts $z_i$ from $s_{i+1}$ alone (the last state of $\tau_i$) will then induce codes with global goal semantics.

**EDDICT Posterior.**  We structure our posterior around goal-conditioned policies $\pi_\theta(a \mid s, z)$, but modified to account for the temporal structure of our prior. We incorporate temporal dependencies between the latent codes in the form of a Markov chain $p(z_i \mid z_{i-1})$ with initial distribution $p(z_0)$. Defining $p(z_0 \mid z_{-1}) := p(z_0)$, we write:

$$p^{\pi_\theta}(\Lambda \mid s_0) = \prod_{i=0}^{M-1} p(z_i \mid z_{i-1}) p^{\pi_\theta}(\tau_i \mid \tau_{i-1}, z_i)$$

We now expand the negative KL-divergence corresponding to this choice of prior, posterior and reverse predictor:

$$-\text{KL}\left[p^{\pi_\theta}(\Lambda \mid s_0) \| p^\mu(\Lambda \mid s_0)\right] = \mathbb{E}_{p^{\pi_\theta}(\Lambda \mid s_0)} \left\{ \sum_{i=0}^{M-1} \log \left[ \frac{q_w^{\text{EDDICT}}(z_i \mid s_{i+1})}{p(z_i \mid z_{i-1})} \frac{p^\mu(\tau_i \mid \tau_{i-1})}{p^{\pi_\theta}(\tau_i \mid \tau_{i-1}, z_i)} \right] \right\}$$

The objective is then obtained by dropping action entropy terms.

$$\mathcal{O}(\theta, w) = \sum_{i=0}^{M-1} \mathbb{E}_{p^{\pi_\theta}(\Lambda_{<i})} \mathbb{E}_{p^{\pi_\theta}(z_i, \tau_i \mid \Lambda_{<i})} \underbrace{\left[ \overbrace{\log q_w^{\text{EDDICT}}(z_i \mid s_{i+1}) - \log p(z_i \mid z_{i-1})}^{\text{ⓒ}} \right]}_{O^{(i)}(\theta, w; z_{i-1}, \tau_{i-1})} \quad (1)$$

The objective thus breaks down as a sum of $M$ terms, defined[5] as $\mathcal{O}^{(i)}(\theta, w) = \mathbb{E}[O^{(i)}(\theta, w; z_{i-1}, \tau_{i-1})]$. It is worth pointing out that in expectation, ⓒ constitutes a valid lower-bound to $\mathcal{I}(z_i; s_{i+1} \mid z_{i-1})$ despite the reverse predictor not conditioning on $z_{i-1}$.

### 3.3  EDDICT Objective

We obtain EDDICT by incorporating (i) greedy optimization, (ii) hindsight correction, and (iii) linear dynamics into the objective of Equation 1.

**Greedy Optimization.**  Define the *effective entropy* as the difference in log-probabilities given by the reverse predictor and the high-level policy over options (cf. ⓒ). As written, the objective aims to maximize the long term sum of effective entropies: concretely, each code $z_i$ should seek to be entropic and discernible from $s_{i+1}$ but also lead to states from which future options are themselves

---

[5]Note that we use $O^{(i)}$ to refer to the $i$-th term of Eq. 1, which is a function of a particular value of $z_{i-1}$ and $\tau_{i-1}$. $\mathcal{O}^{(i)}$ is reserved for the expected value of $O^{(i)}$ under $p^{\pi_\theta}(\Lambda_{<i})$.

discernible. The variance of any return estimator will thus increase with the number of option periods. To avoid this issue, EDDICT optimizes Eq. 1 in a greedy-manner as:

$$\mathcal{O}_{\text{GREEDY}}(\theta, w) = \sum_{i=0}^{M-1} \mathbb{E}_{p^\pi(\Lambda_{<i})} \left[ O^{(i)}(\theta, w; z_{i-1}, \tau_{i-1}) \right], \tag{2}$$

where we have omitted the policy parameters from the sampling distribution $p^\pi(\Lambda_{<i})$, which is thus considered fixed with respect to the optimization process. Concretely, this can be implemented by treating each option period as a pseudo-episode, i.e. using discount factors which are zero on option boundaries as shown in Algorithm 1.

**Hindsight Correction.** Unfortunately, the above objective is rather brittle as the distribution over $z_i$ is conditioned solely on $z_{i-1}$, and ignores the underlying state in which the code is sampled. We can improve on this open-loop formulation by reasoning in hindsight. From Eq. 2, $O^{(i)}$ is computed in expectation under $p^\pi(\Lambda_{<i})$ which includes the joint $p^\pi(z_{i-1}, \tau_{i-1} \mid \Lambda_{<i-1})$. We rewrite this joint as $p^\pi(\tau_{i-1} \mid \Lambda_{<i-1})p^\pi(z_{i-1} \mid \tau_{i-1}) \approx p^\pi(\tau_{i-1} \mid \Lambda_{<i-1})q_w^{\text{EDDICT}}(z_{i-1} \mid s_i)$, since $q_w$ is a variational approximation to the true posterior by construction. Incorporating this approximation to Eq. 2 yields the final objective:

$$\mathcal{O}_{\text{EDDICT}}(\theta, w) = \sum_{i=0}^{M-1} \mathbb{E}_{p^\pi(\Lambda_{<i})} \mathbb{E}_{q_w^{\text{EDDICT}}(z_{i-1}|s_i)} \left[ O^{(i)}(\theta, w; z_{i-1}, \tau_{i-1}) \right]. \tag{3}$$

Concretely, when sampling $z_i \sim p(z_i \mid z_{i-1})$, we thus condition on the code most likely to have yielded state $s_i$, under the reverse predictor. Importantly, this objective induces a cross-entropy term between the target distribution $q_w^{\text{EDDICT}}(z_{i-1} \mid s_i)p(z_i \mid z_{i-1})$ and $q_w^{\text{EDDICT}}(z_i \mid s_{i+1})$. This ensures that predictions made from $s_{i+1}$ are consistent with those from $s_i$, under our latent state dynamics.

**Linear Dynamics** The final piece of the puzzle concerns the choice of code distribution. We cannot employ the VIC strategy of a fixed entropic distribution, since our codes form a Markov chain. We would further like to avoid the full HRL problem, which would require us to have a parameterized high-level policy over options. Choosing an AR(1) process as the conditional code distribution satisfies both of these requirements and we thus set $p(z_i \mid z_{i-1}) = z_{i-1} + \Delta_i$, with $\Delta_i$ sampled from either an isotropic Gaussian or a uniform distribution on the disc. Another useful property of the AR(1) process is that it ensures that the marginal code entropy increases monotonically with each option period (more states visited) while the conditional entropy remains constant (same number of states reachable from any given state), as shown in Fig. 4b. Finally, hard coding the dynamics to be linear, versus learning a parametric policy over codes, naturally imposes an interpretable Euclidian topology in code space, as shown in Fig. 4a.

### 3.4 Algorithm

We now provide a more mechanistic view of EDDICT. Algorithm 1 presents an online version of the algorithm, with details of the distributed setup used in our experiments presented below.

We optimize our objective using a distributed deep reinforcement learning system [14], based on Peng's Q($\lambda$) [37] and $\epsilon$-greedy policies. The system consists of a centralized learner, a replay buffer [32], and a set of distributed workers each interfacing with their own copy of the environment. Given the latest parameter values and current state of the environment $s_i$ (local to each worker), actors sample $z_i$ and generate sub-trajectory $\tau_i$ by executing $\pi(a \mid s, z_i)$ for $K$ steps in the environment. The resulting $(s_i, \Delta_i, \tau_i)$ is then fed back to the replay buffer, from which the learner consumes data to perform off-policy updates. Storing the initiation state $s_i$ and offset $\Delta_i$, allows the learner to recompute the code $z_i$ as required using the most up-to-date version of the reverse predictor. Intrinsic rewards derived from the reverse predictor are similarly computed on the learner.

In practice, the learner maximizes $\mathcal{O}_{\text{EDDICT}}$ by summing two losses. The first implements policy iteration by minimizing the mean-squared error between a target return, computed by Peng's Q($\lambda$) under a target network [32], and the current Q-value estimates. Our greedy optimization procedure yields a single non-zero reward, $\log q_w(z_i \mid s_{i+1})$, which is received upon option termination. The second loss corresponds to the cross-entropy loss of the reverse predictor found in Eq. 3. With $q_w(z \mid s) := \mathcal{N}(f_w(s), 1)$ for some parametric function $f_w$, this amounts to minimizing

---

**Algorithm 1:** EDDICT

---

**Input** : Environment dynamics $P$, initial state $s_0$, policy $\pi_\theta$, code predictor $q_w(z \mid s) \coloneqq \mathcal{N}(f_w(s), \mathbf{1})$, option period $K$, discount $\gamma$, code dimension $d$.

$\boldsymbol{\tau} \leftarrow [s_0], i \leftarrow 0$

**repeat**

    $\Delta_z \sim U(\mathbb{D}^d)$                        *// e.g. uniform over a disc, isotropic normal*

    $z_i \leftarrow f_w(s_{iK}) + \Delta_z$

    **for** $t \leftarrow iK : (i+1)K - 1$ **do**

        $a_t \sim \pi(a|s_t, z_i; \theta)$                   *// parametric or epsilon-greedy*

        $s_{t+1} \sim P(s_{t+1}|s_t, a_t)$

        *// Compute intrinsic rewards. Note: entropy of code distribution is constant under linear dynamics.*

        $r_{t+1} \leftarrow \log q_w(z_i \mid s_{t+1})$ if $t=(i+1)K-1$ else $0$     *// (optional) add entropy rewards.*

        $\gamma_{t+1} \leftarrow 0$ if $t=(i+1)K-1$ else $\gamma$

        $\mathbf{s}_{t+1} \leftarrow [z_i, s_{t+1}]$                  *// augment state with code*

        Append $a_t, \mathbf{s}_{t+1}, r_{t+1}, \gamma_{t+1}$ to $\boldsymbol{\tau}$.

    Update $\theta$ with any reinforcement learning algorithm on the sub-trajectory $\boldsymbol{\tau}$.

    *// Minimize cross-entropy loss from Eq. 3, for linear dynamics and Gaussian reverse predictor.*

    Update $w$ by gradient descent on $\|\Delta_i - (f_w(s_{(i+1)K}) - f_w(s_{iK}))\|_2^2$

    $\boldsymbol{\tau} \leftarrow [s_{(i+1)K}], i \leftarrow i + 1$

---

$\|\Delta_i - (f_w(s_{i+1}) - f_w(s_i))\|_2^2$. This loss is extremely intuitive: we train the reverse predictor such that the inferred latent state from $s_i$, matches the inferred state from $s_{i+1}$ under our latent dynamics. As in [41], we found that an uninformative prior performed best in practice (despite our choice of isotropic Gaussian for the predictor), and thus sample $\Delta_i$ from a uniform distribution on the disc [6].

Concretely, we parameterize the action-value function $Q_\theta(s, a, z)$ as an MLP operating on state embeddings, derived from a ResNet [24], and linear action and code embeddings. In our experiments, the reverse predictor $q_w$ operates on the same state embeddings as the $Q$-function, with gradients from both objectives being backpropagated into the ResNet. Complete details of the architecture can be found in the Appendix.

## 4 Related Work

**Intrinsic Control and Empowerment.** EDDICT can best be thought as incorporating temporal structure into intrinsic control algorithms [19, 15, 1, 21], which build on empowerment [26, 34]. Relative Variational Intrinsic Control (RVIC) [8] also extends an intrinsic control objective, but does so by penalizing codes predictable from a single state, leading to codes representing state-agnostic behaviors. In contrast, the parameterization of our reverse predictor, along with a fixed high-level policy over options, ensure that EDDICT's codes are reachable from the states in which they are sampled while preserving global state semantics.

It is well known that the VIC objective is difficult to train when the code space is large [1]. At a high-level, EDDICT tackles this issue by breaking down this single goal into a sequence of sub-goals. This is orthogonal to the approach of Achiam et al. [1], which increases the number of available options over time. HIDIO [51] proposes an objective similar to ours (discriminator rewards over sub-trajectories, greedy-optimization), but sample options using a state-dependent high-level policy trained to maximize extrinsic rewards over the semi-MDP induced by the low-level policy.

**Skill Discovery and HRL.** The notion of reusable behavior and hierarchy has a long history in the RL literature [e.g. 42]. In comparison to EDDICT existing work can be broadly categorized with respect to the signal that is used for behavior induction and the nature of the learned representation. EDDICT bears similarity to *unsupervised* skill discovery methods that induce behavior in the absence of external rewards usually for the use in downstream tasks, including [16, 36]. Other approaches learn skills or behavior representations from demonstrations provided by humans or expert policies [e.g.

---

[6]Our variational bound is looser as a result, since our variational posterior is not matched to the prior. Improving the modeling assumptions of the reverse predictor, e.g. by using a truncated Gaussian, is left for future work.

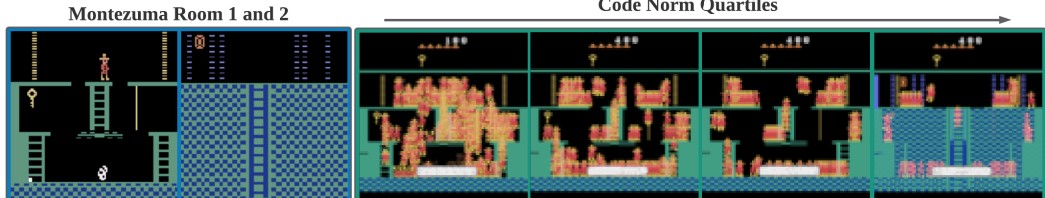

**Montezuma Room 1 and 2**   **Code Norm Quartiles**

Figure 2

Figure 3: **Montezuma's revenge. (left)** Typical observations from the first (left) and second (right) rooms. **(right)** Observations from a trained EDDICT agent, sorted by $L2$-norm of corresponding code and aggregated into quartiles. Images are generated by taking the pixel-wise maximum

17, 31, 38], while optimizing the reward for one or multiple tasks [e.g. 6, 25, 23, 44, 49, 29, 18], or via subgoals that are associated with explicit rewards in a predefined [e.g. 28, 35] or learned space [e.g. 46, 47].

**Methodology.** Auxiliary variables have a long history in variational inference [3, 30], as a way to obtain more expressive posteriors and serve a similar purpose in the context of EDDICT/VIC. Tirumala et al. [45] incorporates an AR(1) process in the context of HRL and skill transfer, but did so within the prior which served to regularize a set of task-specific high-level policies. In contrast, the AR(1) process in EDDICT ensures that the high-level policy samples goals which are reachable from the current state. AR(1) processes over latent temporal sequences have also been used to prevent "posterior collapse" in VAEs with powerful autoregressive decoders [39], an analogous phenomenon to "option collapse" in HRL. Hindsight reasoning has seen a multitude of applications in reinforcement learning, improving credit assignment [22], training of goal-conditioned policies in sparse reward settings [5] and off-policy learning of options in HRL [50].

## 5 Experiments

Here we evaluate EDDICT's learned representations and behavior, and contrast them to prior work in the space of intrinsic control (or skill discovery) methods. We assess the learned representations qualitatively by looking at how well they correspond to privileged information known to be relevant to down stream tasks. Namely, the state dimensions given in the DeepMind Control Suite [43] and the avatar coordinates in the Atari Learning Environment (ALE) [9]. We stress that this privileged information is not used during training in any way, with reverse predictors operating on the same input as the Q-function.

The quality of learned behaviors is measured in terms of exploration; we posit that EDDICT explores in the space of controllable outcomes, and that this style of exploration results in reaching many states of interest. To assess this quantitatively, we compare unsupervised behavior policies in terms of reward achievement on the Atari game *Montezuma's Revenge*, which is known to require sophisticated exploration in order to progress. Additionally, we look at the number of unique states visited per episode using privileged environment information (i.e. the underlying RAM states in ALE), as this is a proxy for state coverage that is agnostic to the specific reward function of the game [4]. To look specifically at the claim that EDDICT explores the *controllable* states, we also measure an estimate of the mutual information between the marginal code distribution and the marginal state distribution.

We consider the following baselines for evaluation: VIC [19], RVIC [8] and an ablated version of EDDICT. VIC refers to a scalable variant introduced in [8], that uses a fixed Categorical distribution over 50 outcomes. In the EDDICT ablation (EDDICT–$\Delta$) the code proposal mechanism is simplified by substituting $\dot{z}_{i+1} := \Delta_i$ for EDDICT's $z_{i+1} := z_i + \Delta_i$. Note that the reverse predictor remains unchanged, and thus tries to predict $\Delta_i$ directly from $s_{i+1}$. All algorithms were implemented in the same codebase and thus share the same network architecture and reinforcement learning method.

For the results on *Montezuma's Revenge*, we further include results for a $Q(\lambda)$ agent trained to maximize the game score (which other methods do not have access to), again matched in terms of network architecture.

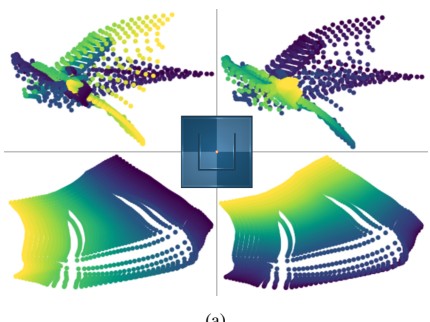
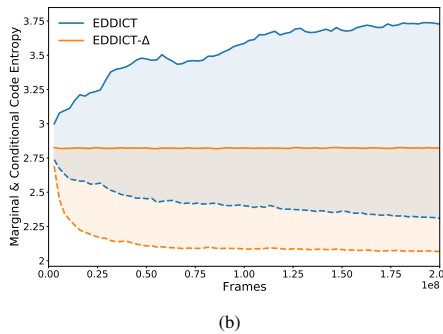

(a)                                              (b)

Figure 4: **(a) Point Mass.** 2D codes colored by ground truth x (left) and y (right) coordinates of the point-mass on a version of the environment with a 'U' shaped wall. (top) EDDICT-$\Delta$ (bottom) EDDICT. **(b) *Montezuma's Revenge*.** Estimated marginal code entropy $\mathcal{H}[z_i]$ (solid) and conditional entropy $\mathcal{H}[z_i \mid s_{i+1}]$ (dashed). Despite codes being less predictable, EDDICT achieves higher mutual information $\mathcal{I}_p(z_i; s_{i+1})$, as measured under the marginal code distribution (shaded).

## 5.1 Codes as Representations of State

**Codes as state representations**   To illustrate EDDICT's ability to map codes to the controllable aspects of the environment, we have trained EDDICT on a simple continuous control task from the DeepMind Control Suite [43]. This environment comes with a set of ground truth state dimensions required to compute the dynamics, a subset of which are under the agent's control. We can thus allow EDDICT to train from raw pixels, and then evaluate the correspondence between the code values and ground truth state dimensions.

Shown in Figure 4a, is the state representation learned on a version of the Control Suite point mass domain which is modified to include 3 obstacle walls in the shape of a "U" to increase the difficulty and exploration requirements of the domain. EDDICT successfully recovers the ground truth coordinates of the point mass position (under the agent's control), but not the target position (randomly set per episode and not under the agent's control). This property of not representing what can not be controllable effectively solves the well known "noisy TV problem" [10].

Standard intrinsic control methods also have this property of only representing the controllable, but they lack any incentive to represent states unreachable in a single unroll as being distinct. This is clearly demonstrated by our ablation's performance, wherein the relationship between nearby states is much more tenuous.

**Code norm as difficulty**   Since our desired latent dynamics consist of a sum of zero centered IID variables, the marginal code distribution will be also centered at zero, with the probability density dissipating as a function of the code norm. Assuming EDDICT manages to form a mapping between latent codes and states, this implies that less frequently visited states will have a higher code norm.

We test this hypothesis empirically, by training EDDICT on *Montezuma's Revenge*. In this game, the agent is represented by an avatar who can move locally around the screen, but who can easily die. When this occurs, the avatar is reborn on a platform in the middle of the screen. This means that by construction, states where the avatar is near the center of the screen as easier to reach than those farther out. Our hypothesis would thus suggest that EDDICT should assign center states with low norm codes and more peripheral states with high norm codes. As Figure 2 shows, this is exactly what happens in practice. One interesting subtlety is that the game actually contains several rooms, each with a different background. These are exceedingly hard to reach and, as expected, this results in EDDICT assigns these states the highest code norms of all.

## 5.2 Control to Explore

EDDICT's representations only tell half of the story. Since codes represent states, and the marginal code entropy increases monotonically, this suggests EDDICT's fixed high-level policy should result in exploratory behavior. But unlike most traditional work on the exploration problem, EDDICT focuses its exploration only on what it can control.

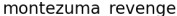

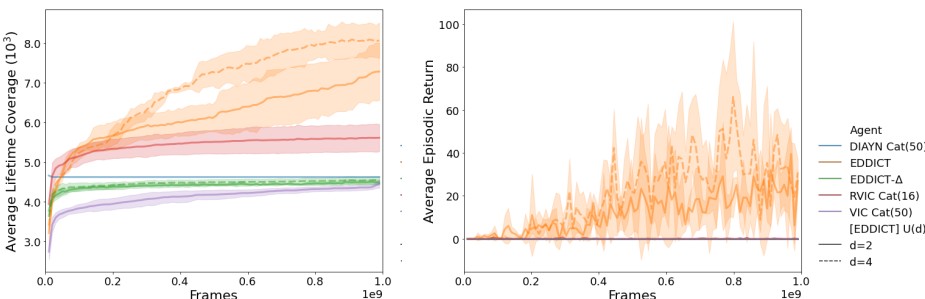

Figure 5: Comparing exploratory behavior in *Montezuma's Revenge*. **(right)** Game score. All methods except $Q(\lambda)$ did not have access to this during training. **(left)** Number of unique avatar positions visited. This is a proxy for coverage of the controllable states.

**Measuring control**    Intrinsic control methods can measure their control over the environment in terms of the mutual information between a code and its downstream effects of the environment. For episodic or resampled but independent codes, this can be estimated straightforwardly as the effective code entropy (cf. ⓒ, Eq. 1) averaged over a mini-batch. EDDICT however defines a Markov chain over codes, and thus requires us to compute entropy over the *marginal code distribution*. To do so, we fit an isotropic Gaussian to all of the codes in the batch, yielding mean and variance estimates $\hat{\mu}$ and $\hat{\sigma}^2$. Our final estimate of $\mathcal{I}_{q_w}(z_i; s_{i+1})$ is then $\mathcal{H}\big[\mathcal{N}(\hat{\mu}, \hat{\sigma}^2)\big]$, plus the average log-prediction reward $\log q_w(z_i \mid s_{i+1})$ over the batch.

Since this is a lower bound, it is not an unbiased estimate, but the relative values should still be meaningful when comparing models of the same architecture. This metric thus allows us to answer the question: does EDDICT control the environment to a greater degree than its fixed code distribution equivalent? As Figure 4b shows, this is very much the case. Interestingly, this is true despite the codes in general being less predictable; the extra entropy from the sequential sampling more than makes up for it.

**Exploring what matters**    In order to evaluate EDDICT's behavior policy, we must designate a proxy metric for exploration quality. For Atari 2600, game score is an obvious candidate. But since the methods under consideration learn without access to score or episode boundaries, this metric is sometimes quite noisy. To give a more complete view of exploratory behavior, we also include two 'coverage' metrics which counts the number of unique RAM states the agent visits per episode and over its lifetime. Using the information given in [4], we only count the RAM states corresponding to the controllable avatar. We evaluate EDDICT on these 3 metrics across 6 amenable games, and as shown in Table 1, we find that in the majority of cases EDDICT outperforms RVIC, VIC and its ablation on one or more metrics. RVIC was the most competitive method, and suggests that global codes are not the only way for intrinsic control methods to yield exploratory behavior.

Of particular interest is *Montezuma's Revenge*, one of the hardest exploration games, as attested by the numerous reinforcement learning papers that fail to receive non-zero scores (e.g. [33, 14]). As shown in Figure 5, EDDICT outperforms other intrinsic control methods by a wide margin.

For additional context, we provide a broader set of baseline results on Montezuma's Revenge in Table 2 of the Supplemental. These span entropy or curiosity-based algorithms which learn a single unified policy, instead of the code-conditional policies recovered by algorithms in the VIC family, with Never-Give Up [7] and Random Network Distillation [11] greatly outperforming EDDICT on the metric of average episodic reward.

## 6   Limitations and Discussion

Endowing agents with the ability to master the environment is an important step towards more general purpose agents, as it allows learning in any circumstance without any requirement of a task specific

| Game | DIAYN | RVIC Cat(16) | EDDICT | EDDICT-$\Delta$ |
|---|---|---|---|---|
| Berzerk | **1.75**, 24.8, 365 | 0.156, 31.3, 138, | 0.382, **61.3**, 477, | 0.562, 10.6, **584** |
| Hero | 1.82, 24.1, **1.35k** | **2.58**, **35.2**, 805, | 2.1, 32.5, **1.34k**, | 0.856, 20.7, 68.7 |
| Montezuma | 0.379, 4.61, 0 | 0.577, 5.61, 0, | **1.17**, **8.07**, **30.9**, | 0.674, 4.54, 0 |
| Ms. Pacman | **0.475**, 1.72, **652** | 0.288, 1.72, 397, | 0.349, 1.72, 587, | 0.262, 1.72, 360 |
| Private Eye | 4.61, **87**, 1.54k | **5.41**, 85.2, 886, | 4.86, **86.8**, 1.07k, | 4.01, 74.5, -43.6 |
| Seaquest | 0.346, 10.9, 18.7 | 1.27, 10.9, 143, | **1.91**, 10.9, **400**, | 1.56, 10.9, 238 |

Table 1: Results on 6 Atari games at 1B frames. Each tuple A,B,C represents mean of: (A) Episodic Coverage ($\cdot 10^3$) (B) Lifetime Coverage ($\cdot 10^3$) (C) Average return. For EDDICT-based agents, we pick the best metric across code sizes. Metrics which are best across agents, based on mean performance over 3 seeds, are shown in bold. In cases where there is a tie across all methods for a metric, none are bolded. The full set of results, including VIC, random policy baselines, scores with standard deviations and training curves can be found in the supplemental material.

reward function. EDDICT can both explore and control the environment by learning latent codes that make sense of states in a globally consistent coordinate system. But in terms of pure exploration, EDDICT falls short of state of the art methods that learn a single policy (e.g. [12, 11]). Understanding what these advancements mean for learning code-conditional policies is a promising future direction.

Additive dynamics can not capture important aspects of some environments that we might wish for our agents to represent, such as dynamics that are irreversible or state-dependent. Ideally, general purpose function approximators (e.g. neural networks) could be used to specify more general dynamics, but how to make such learning tractable while preserving the advantageous properties of EDDICT remains an important open question.

In addition to its inherent merits for environment exploration and manipulation, EDDICT's novel state to code mapping and code transition function could be used to aid local planning, or could serve as a compact representation on top of which to learn policies, or a good state similarity metric for goal-based RL, or aid in many other unlisted tasks. All of these directions are left as future work.

# 7 Societal Impact

Unsupervised reinforcement learning in general, and intrinsic control methods in particular, are far from being commercialized due to their insufficient data efficiency and lack of validation in real world environments. However, when this is no longer the case, these methods could significantly reduce the human cost of setting up systems that interact with humans (e.g. robotics), as these methods limit the need for handcrafted reward functions and the collection of human preferences. But this benefit comes with a cost to interpretability and safety. The information theoretic objectives of the methods lead to behavior that can be very hard to predict a priori (e.g. what does 'controlling your environment' look like?). Furthermore, safety constraints might be harder to specify in the absence of a closed-form reward function. As these methods mature, the emphasis should shift from raw performance to a more nuanced approach that addresses these societal concerns head on.

# 8 Acknowledgements

We would like to thank Stephen Spencer for providing engineering support. We further thank Yury Sulsky and Arturo Bajuelos, who contributed to results found in Appendix B, showing how EDDICT can be incorporated into the exploration policy of standard RL agents.

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
