# Entropic Desired Dynamics for Intrinsic Control: Supplemental Material

**Steven Hansen**[*]
DeepMind

**Guillaume Desjardins**
DeepMind

**Kate Baumli**
DeepMind

**David Warde-Farley**
DeepMind

**Nicolas Heess**
DeepMind

**Simon Osindero**
DeepMind

**Volodymyr Mnih**
DeepMind

## A   Improving Exploration for Standard RL

While the focus of this work has been on unsupervised evaluation, here we provide a proof-of-concept that EDDICT can aid in the achievement of task rewards via structured exploration.

To do so, we train EDDICT alongside a standard policy maximizing task rewards, which we refer to as the *task policy*. While the EDDICT training procedure remains unchanged, experience for the latter is generated by a behavior policy which randomly switches between EDDICT's policy and the task policy at regular intervals (every 20 steps). The motivation is similar to recent work [8] on temporally extended $\epsilon$-greedy: temporally coherent exploration can more rapidly cover the state space. Two separate networks were used to instantiate EDDICT and the task policy, so any potential benefits must arise from improved exploration rather than e.g. representation regularization via an auxiliary objective. We stress that this is likely to be a sub-par utilization of EDDICT, since this neglects many of it's unique characteristics (e.g. learning a smooth latent state representation). We have restricted ourselves here to avoid additional complexity and focus on evaluating the exploratory benefits of EDDICT in the most straight-forward way possible.

Figure 1 demonstrates that having EDDICT contribute to the behavior policy in this way is beneficial to performance on *Montezuma's Revenge*. While this is not close to the state-of-the-art in general (c.f. [2]), it is competitive in the data-limited regime [15].

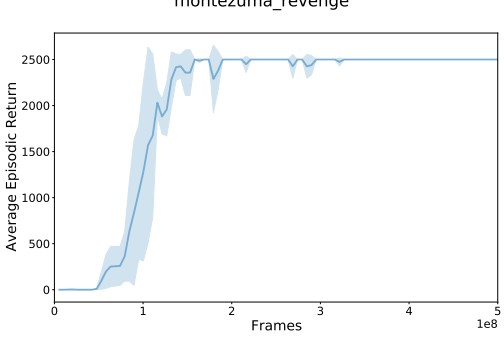

Figure 1: **Average Episodic Return.** Training EDDICT alongside a standard RL agent maximizing extrinsic rewards, and using the former for exploration, yields increased score achievement.

---

[*]Correspondence to stevenhansen@deepmind.com

35th Conference on Neural Information Processing Systems (NeurIPS 2021), Sydney, Australia.

## B  Action Entropy

The version of EDDICT reported in the main text used an $\epsilon$-greedy behavior policy trained with Peng's $Q(\lambda)$ [16]. This is in keeping with prior work on using intrinsic control methods in Atari [12]. But much of the literature uses a parametric behavior policy with action entropy as part of the objective (e.g. [10]). In this section, we run EDDICT using soft Q-learning in order to similarly incorporate action entropy [11, 18].

Figure 2 shows the effect of action entropy on exploratory behavior in Montezuma's Revenge. While it appears there is some benefit to including action entropy in the EDDICT objective, the results are not particularly significant. However, this does at least suggest that action entropy is not actively harmful, and need not be avoided even in environments with irreversible dynamics like Atari.

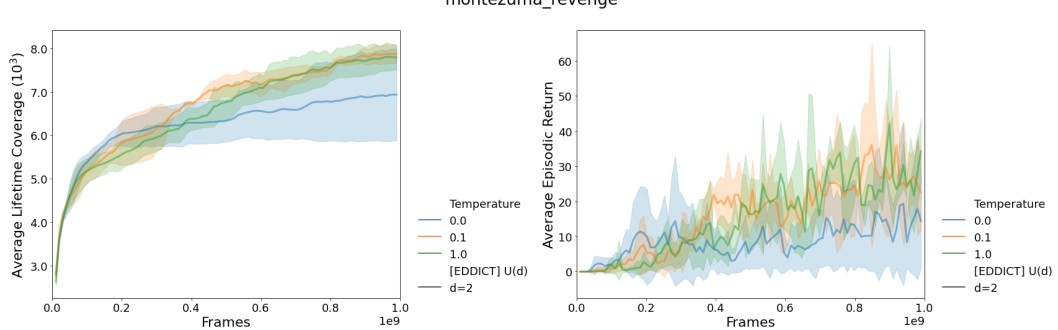

Figure 2: **EDDICT with action entropy** Training EDDICT while varying the weight of the action entropy term in the objective function. Zero corresponds to reverting back to Peng's $Q(\lambda)$. **(right)** Game score. **(left)** Number of unique avatar positions visited. This is a proxy for coverage of the controllable states.

## C  Full Atari Training Curves

Full training curves across all 6 Atari games are shown in Fig. 3, including the random policy baseline. Table 3 is a more in-depth version of Table 1 found in the main paper, modified to include standard deviation estimated across 10 seeds for RVIC, EDDICT and EDDICT-$\Delta$.

## D  DIAYN Hyper-Parameter Sweep

Diversity is All You Need (DIAYN) is included as a baseline despite our experimental setup differing significantly from that of the original paper [10]. The environment is not reset after each skill execution, and the environment is Atari rather than continuous control. These differences motivated using no action entropy and a shorter skill duration. To ensure this didn't hamper performance, we ran a grid search over these two hyper-parameters.

As shown in Figure 4, neither of these hyper-parameter had a significant impact, even when varied across multiple orders of magnitude.

## E  Coverage Calculation

At each state visited by the agent evaluator during training, the agent's state (consisting of the avatar's x and y coordinates within the frame, and potentially also the room number in games with more than one frame in which the agent can move, such as the different rooms in Montezuma's Revenge) is extracted from the environment's RAM state using the RAM annotations provided by [1]. To compute coverage, the evaluator counts all unique (x, y, [frame number]) states visited either within its lifetime (for lifetime coverage) or since the last episode reset (for episodic coverage).

We note that the evaluator runs in parallel to and is not rate limited with respect to the learner or other actors, and therefore, lifetime coverage could increase more quickly for some methods than others

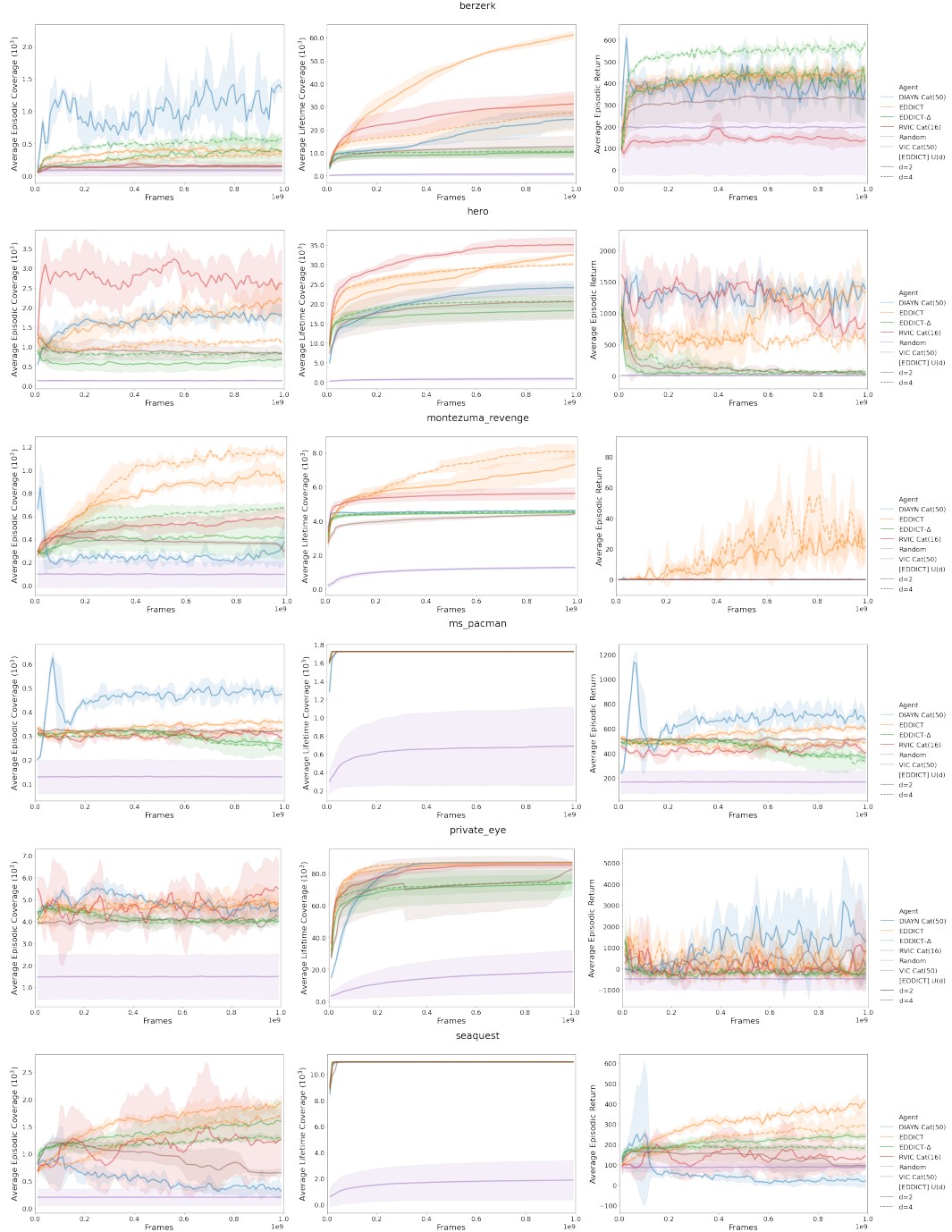

Figure 3: **Full Atari Training Curves** Each row shows (from left to right) Episodic Coverage ($\cdot 10^3$), Lifetime Coverage ($\cdot 10^3$), and Average return for a specific game. These results correspond to those of Table 1 in the main text.

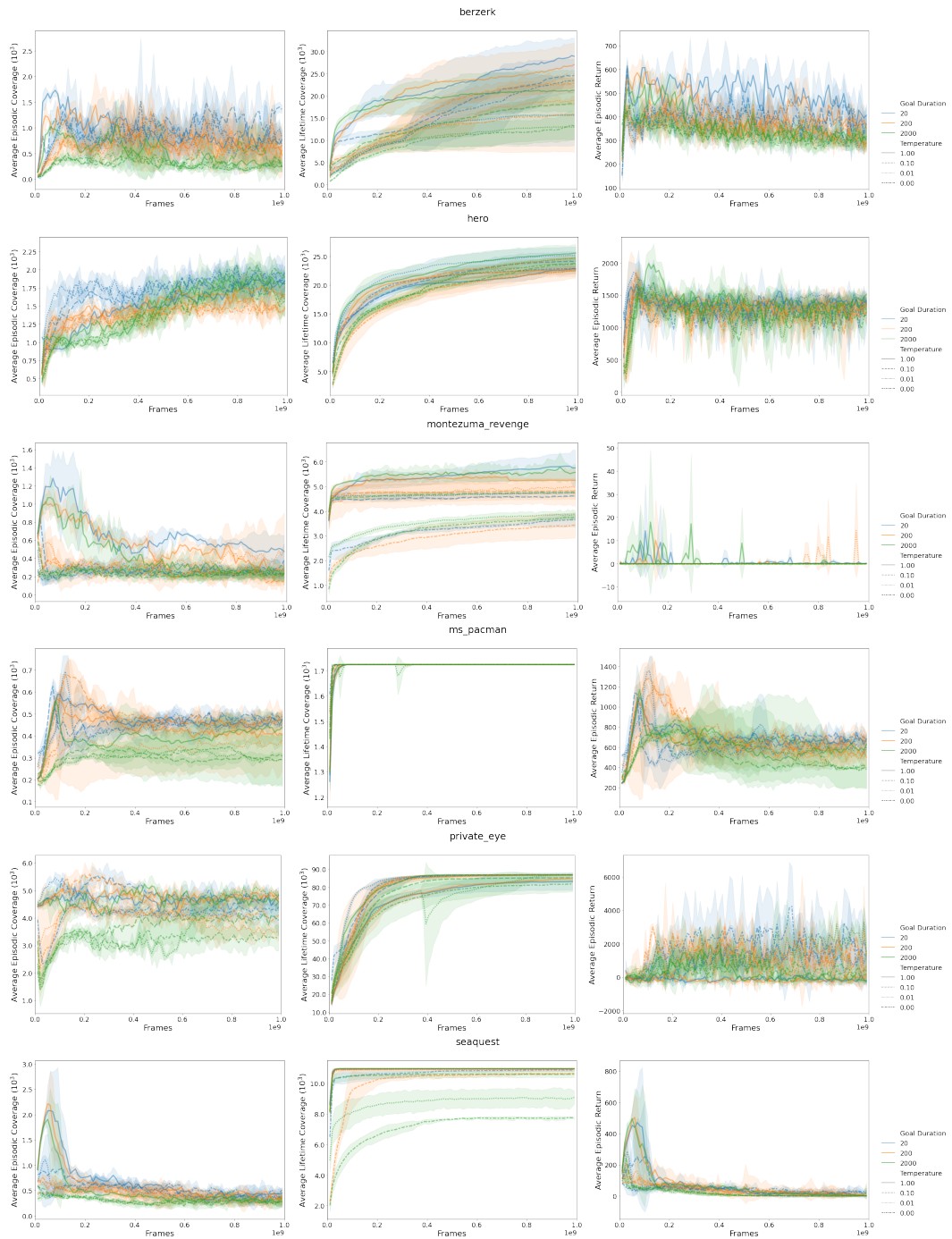

Figure 4: **Full Atari Training Curves for DIAYN grid-search** Each row shows (from left to right) Episodic Coverage ($\cdot 10^3$), Lifetime Coverage ($\cdot 10^3$), and Average return for a specific game.

due to the relative speed of the evaluator compared to the number of actor frames. Therefore, the speed of takeoff for the lifetime coverage metric in Figure 3 should be devalued in favor of paying attention to asymptotic performance. The full curves are included for completeness.

## F  Compute Requirements

The compute cluster we performed experiments on is heterogenous, and has features such as host-sharing, adaptive load-balancing, etc. It is therefore hard to give precise details regarding compute resources – however, the following is a best-guess estimate.

A full experimental training run lasted 4 days on average. Our distributed reinforcement learning setup [9] used 100 CPU actors and a single V100 GPU learner. Thus, we required approximately 9600 CPU hours and 96 V100 GPU hours per seed, with 3 seeds and 7 conditions per game and with followup experiments requiring 10 seeds for 3 conditions. No significant hyper-parameter tuning was required, as the values were inherited from previous work on VIC and general reinforcement learning in the same setup. For the only EDDICT specific hyper-parameter, the latent dimensionality, only 2 values were considered, which account for 4 of the 6 conditions mentioned above. The results were dominated by the compute required for the set of six Atari games i.e. the cost of the qualitative results was not significant. By the above approximation, these experiments required in total 489,600 CPU hours and 4,896 V100 GPU hours.

It is worth remembering that the above is likely quite a loose upper-bound, as this estimate assumes 100 percent up time, which is far from the truth given the host-sharing and load-balancing involved in our setup. Additionally, V100 GPUs were chosen based on what was on hand; our models are small enough to fit on much cheaper cards without much slowdown.

## G  Architectural Details and Hyperparameters

Architectural details and hyper-parameters used in the experiments are shown in Table 1.

| Torso | IMPALA Torso [9] |
|---|---|
| Head hidden size | 512 |
| Number of actors | 100 |
| Batch size | 64 |
| Steps per option | 20 |
| Unroll length | 20 |
| Actor update period | 100 |
| Latent code dimension | [2,4] |
| Replay buffer size | $10^6$ unrolls |
| Optimizer | Adam [13] |
| learning rate | $2 * 10^{-4}$ |
| Adam $\epsilon$ | $10^{-2}$ |
| RL algorithm | $Q(\lambda)$ [17] |
| $\lambda$ | 0.7 |
| discount $\gamma$ | 0.99 |
| Target update period | 100 |

Table 1: A table of the hyperparameters for all experiments. Hyperparameter values are shared across all ALE Atari games and the DeepMind Control Suite point mass task. All baselines and ablations shared all applicable hyper-parameters, which were tuned for standard reinforcement learning rather than EDDICT specifically. One exception was the RVIC baseline, whose hyperparameters were copied from Baumli et al. [4].

## H  Unsupervised Baselines for *Montezuma's Revenge*

Here we present previously published results on unsupervised Montezuma's Revenge that cover a wide range of methods that can broadly thought of as being novelty-seeking. This highlights the

**Montezuma Room 1,2, and 3**

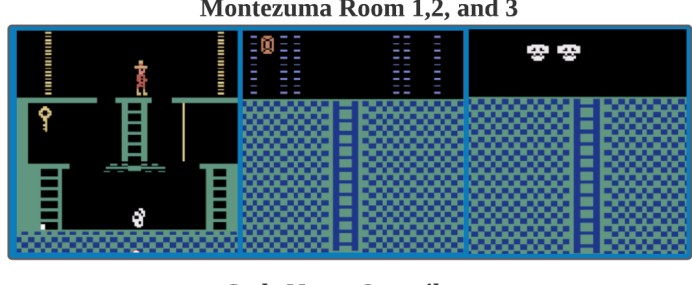

**Code Norm Quartiles**

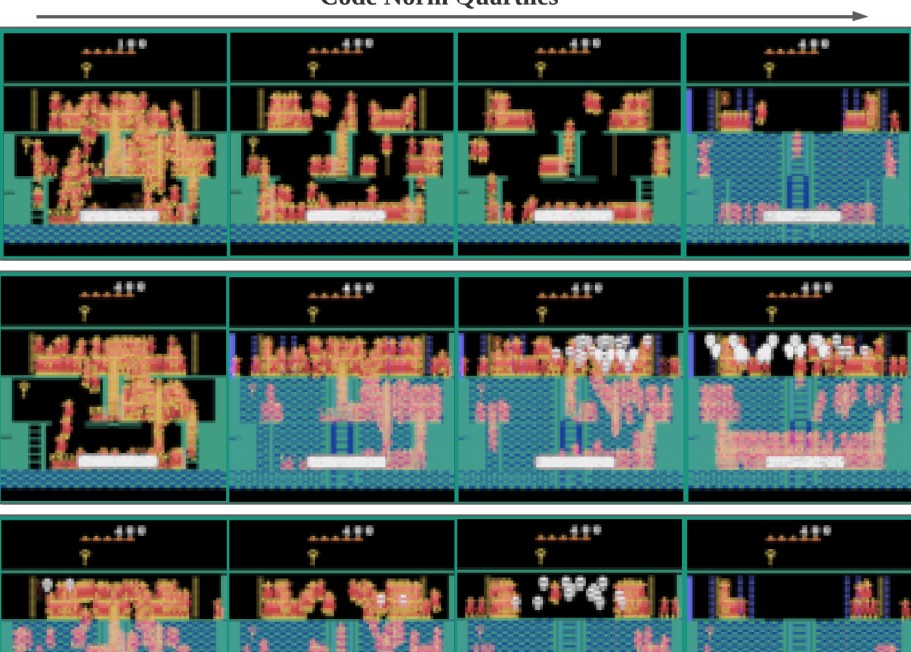

Figure 5: **Expanded Code Norm Results** Additional seeds (bottom rows) added to show the qualitative variability of EDDICT representations for different training runs. These differ in their preference for going to room 2 vs room 3, but all demonstrate the main effect of more difficult locations corresponding to codes with a higher norm.

relative merits of different methods. As mentioned in the discussion, learning a single policy appears to currently be more scalable in terms of just producing exploratory behavior, and Table 2 make this concrete. That said, we note that intrinsic control / latent code methods have a unique set of use cases compared to other novelty-based approaches, due to learning a family of policies (i.e. code conditional) instead of a single policy.

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

| Method | Score | Env. Frames | Notes |
|---|---|---|---|
| EDDICT (this work) | [20-80] | 1B | Figure 3 |
| Active Pre-Training (APT) [14] | 0.2 | 250M | Table 10 |
| Large Scale Curiosity, Inverse Dynamics [5] | 0 | 400M | Figure 2 |
| Large Scale Curiosity, Random CNN [5] | [150, 400] | 400M | Figure 2 |
| Never Give-Up (NGU) [3] | 2.6k | 1B | Figure 1 [7] |
| Random Network Distillation (RND) [6] | 6.7k | 2B | Section 3.1 |

Table 2: Average episodic score on *Montezuma's Revenge* for EDDICT and other baseline methods which do not recover goal-conditioned policies. None of the above methods have access to the extrinsic reward during training. Due to a lack of standardized evaluation, we report results as found in the literature, for a varying number of environment training frames.

strategies. In *International Conference on Learning Representations*, 2020. URL `https://openreview.net/forum?id=Sye57xStvB`.

[4] K. Baumli, D. Warde-Farley, S. Hansen, and V. Mnih. Relative variational intrinsic control, 2020.

[5] Y. Burda, H. Edwards, D. Pathak, A. Storkey, T. Darrell, and A. A. Efros. Large-scale study of curiosity-driven learning. *arXiv preprint arXiv:1808.04355*, 2018.

[6] Y. Burda, H. Edwards, A. Storkey, and O. Klimov. Exploration by random network distillation. *arXiv preprint arXiv:1810.12894*, 2018.

[7] V. Campos, P. Sprechmann, S. S. Hansen, A. Barreto, S. Kapturowski, A. Vitvitskyi, A. P. Badia, and C. Blundell. Beyond fine-tuning: Transferring behavior in reinforcement learning. In *ICML 2021 Workshop on Unsupervised Reinforcement Learning*, 2021.

[8] W. Dabney, G. Ostrovski, and A. Barreto. Temporally-extended {\epsilon}-greedy exploration. *arXiv preprint arXiv:2006.01782*, 2020.

[9] L. Espeholt, H. Soyer, R. Munos, K. Simonyan, V. Mnih, T. Ward, Y. Doron, V. Firoiu, T. Harley, I. Dunning, et al. Impala: Scalable distributed deep-rl with importance weighted actor-learner architectures. *arXiv preprint arXiv:1802.01561*, 2018.

[10] B. Eysenbach, A. Gupta, J. Ibarz, and S. Levine. Diversity is all you need: Learning skills without a reward function, 2018. URL `http://arxiv.org/abs/1802.06070`.

[11] T. Haarnoja, H. Tang, P. Abbeel, and S. Levine. Reinforcement learning with deep energy-based policies. In *Proceedings of the International Conference on Machine Learning (ICML)*, pages 1352–1361, 2017.

[12] S. Hansen, W. Dabney, A. Barreto, T. Van de Wiele, D. Warde-Farley, and V. Mnih. Fast task inference with variational intrinsic successor features. *arXiv preprint arXiv:1906.05030*, 2019.

[13] D. Kingma and J. Ba. Adam: A method for stochastic optimization. *arXiv preprint arXiv:1412.6980*, 2014.

[14] H. Liu and P. Abbeel. Unsupervised active pre-training for reinforcement learning, 2021. URL `https://openreview.net/forum?id=cvNYovr16SB`.

[15] M. C. Machado, M. G. Bellemare, and M. Bowling. Count-based exploration with the successor representation. In *Proceedings of the AAAI Conference on Artificial Intelligence*, volume 34, pages 5125–5133, 2020.

[16] J. Peng and R. J. Williams. Efficient learning and planning within the Dyna framework. *Adaptive Behavior*, 2:437–454, 1993.

[17] J. Peng and R. J. Williams. Incremental multi-step Q-learning. *Machine Learning*, 22:283–290, 1996.

[18] J. Schulman, X. Chen, and P. Abbeel. Equivalence between policy gradients and soft q-learning. *arXiv preprint arXiv:1704.06440*, 2017.

| Game | RVIC Cat(16) | EDDICT | EDDICT-Δ |
|---|---|---|---|
| Berzerk | 0.16±0.017, 37.6±3.8, 151±15, | 0.397±0.076, **71±2.3**, 453±29, | 0.553±0.16, 14.4±1.9, 546±74, |
| Hero | 2.82±0.46, 38.8±1.9, 1.22e+03±4.8e+02, | 2.18±0.2, 37.6±2.2, 1.28e+03±1.3e+02, | 0.991±0.049, 24.6±0.94, 90.1±20, |
| Montezuma | 0.684±0.095, 5.9±0.25, 0.00314±0, | **0.944±0.17, 7.89±0.65, 25.2±5.9**, | 0.684±0.059, 4.72±0.072, 0±0, |
| Ms. Pacman | 0.283±0.02, 1.72±0, 391±51, | **0.355±0.033**, 1.72±0, **602±83**, | 0.269±0.025, 1.72±0, 377±67, |
| Private Eye | 3.4±0.74, 85.1±1.7, -63±2.4e+02, | 4.74±0.17, 87.2±0.021, 73.3±1.1e+03, | 4.38±0.32, 85.1±2.1, -245±86, |
| Seaquest | 0.67±0.16, 10.9±0, 101±46, | 1.82±0.16, 10.9±0, **389±32**, | 1.52±0.16, 10.9±0, 232±18, |

Table 3: Results on 6 Atari games with 10 seeds per condition at 1B frames. Each tuple A,B,C represents a confidence interval for: (A) Episodic Coverage ($\cdot 10^3$) (B) Lifetime Coverage ($\cdot 10^3$) (C) Average return. For EDDICT-based agents, we pick the best metric across code sizes. Metrics which are best across agents without overlapping confidence intervals are shown in bold.