# OpenReview forum: "Entropic Desired Dynamics for Intrinsic Control"
_NeurIPS.cc/2021/Conference — NeurIPS 2021 Poster_

### Official Review · Reviewer_gr67 · 2021-07-15

**Rating:** 7
**Confidence:** 3

**Summary:**

This paper presents a skill discovery method based on variational intrinsic control. The main idea is to allow the skill codes to represent skills that can be reached at longer time horizons than the skill period through the use of Markovian dynamics of skill variables. The method is derived in terms of a variational inference objective over episodes applied to a prior distribution with a fixed policy and learned skill inference, and a posterior distribution with learned policy and randomized linear skill dynamics. Off-policy RL is used to optimize the resulting intrinsic objective. The paper presents experiments that compare the method to others in terms of coverage metrics and extrinsic reward metrics. The experiments show that the method exhibits generally superior performance in terms of coverage across the investigated set of Atari environments, and qualitatively, that the latent codes exhibit some structure correlated with the spatial structure of the environment.

**Limitations And Societal Impact:**

The paper mentions that the method falls short of methods e.g. [10,11], yet these methods are not compared to. The paper needs to include more context for this statement (numbers!) as well as justify why more detailed comparison is not needed, or why the proposed method is still significant.

The societal impact statement is embedded in the checklist -- instead, I think it should appear either in the main text or in the supplementary material.

**Main Review:**

The motivation of the paper makes sense -- the second paragraph did a good job of making the goals of the paper clear. The resulting method appears to be original. The results demonstrate that the method achieves better coverage than other methods, but its not clear how significant this result is in terms of 'downstream' tasks that rely on large coverage. The main concern I have with this paper is the clarity in the mathematical presentation. This concern stems from a variety of points, ordered from highest to lowest priority:

**Concerns w.r.t. clarity and correctness of the method:**
- The presentation in Sec 3.1 needs to be improved. L81 says "we can show that a regularized version of VIC...", yet fails to present or introduce the unregularized version of VIC.
- As far as I can tell, Eq (3) is equivalent to Eq (2), yet L133 describes Eq (3) as an alternative way to optimize Eq (3). Relatedly, (3) is objective function, not an optimization problem, so the presentation of (3) doesn't function as a description of an optimization problem, like how the paper text describes it.
- L117 $\mathcal O_{\text{VIC}}$ is undefined.
- L134 and L138: $p^\pi(\Lambda_{< i})$ was never explicitly defined. Thus, it's not clear on L138 that it includes the $p^\pi(z_{i-1}, \tau_{i-1} | \Lambda_{< i - 1})$ (its not clear that the full joint decomposes in this way). I think the definition may be implicit in the posterior definition, even if this is true, this needs to be clarified.
- Eq 4 doesn't quite make sense -- the final term in the joint needs to be removed from the outer expectation, otherwise, there are two expectations over distributions of $z_{i-1}$.
- More motivation is needed in L73-75 for why it is useful to view the learning procedure in the context of [19,12,26], rather than simply as variational inference. This doesn't appear to offer the method any clear technical motivation or starting point. This paragraph seems unnecessary -- the main point is the construction of the prior and posterior for VI.

**Minor concerns:**
- L256 "This is clearly demonstrated by our ablation’s performance, wherein the relationship between nearby states is much more tenuous." I don't see where the ablation's relationship between nearby states is investigated. E.g. I expected them to appear in Fig 4a, but instead, only the main method and VIC are used (not the ablation, EDDICT-$\Delta$).
- Sec 2: The paper should probably use sequence notation for trajectories instead of set notation, because braces commonly denote absence of ordering (whereas parentheses commonly denote presence of ordering).
- The brace sizes on the first line of Eq. 2 are mismatched.
- The second bracket in $i \in [0, M]$ (below L110) is reversed.
- The Fig1. caption is missing a description of what the diamond denotes in Fig 1d.
- L73 typo in Intrinsic
- Below L114 "between the latent codes in the form of a Markov chain" -> "between the latent codes in the form of a Markov chain with dynamics"
- Fig 4b should denote some of the MI values numerically (e.g. those at 0 and 2e8).


**Time Spent Reviewing:**

4

---

> ### Author Response · Authors · 2021-08-11
> **Response to Reviewer gr67**
>
> We thank the reviewer for the detailed feedback and positive comments regarding motivation, originality and recognizing the efficacy of the proposed method in improving coverage.
>
> Regarding the significance of the result on 'downstream task'. We strongly believe that learning a rich set of skills which are tailored to the particular environment dynamics, but agnostic to the reward function is key to improving data efficiency in reinforcement learning. This may be the ultimate "downstream task". This paper takes one step in this direction, improving on SoTA intrinsic control methods by introducing temporally correlated skills which improve coverage and improved representations. While we do believe the resulting options could be used as part of a full HRL agent (as done in DIAYN and RVIC), we believe this is best left as a follow-up paper. We do have reason to be optimistic however: by co-training an exploration policy with EDDICT, a standard RL agent maximizing rewards can achieve up to [2.5k results on Montezuma](https://imgur.com/a/0Dspd2a).
>
> While we agree there are some notational issues, we are happy to see this did not preclude proper understanding of the method. We thus hope to have an opportunity to incorporate your feedback in this regard, and improve the notation and clarity of the paper for future revisions. We address your other points in more detail below:
>
> ## [major]
> * The unregularized objective of VIC corresponds to \textcircled{a}, with the particular choice of reverse predictor specified on line 88. We will make this more explicit, along with $\mathcal{O}_VIC}$.
> * The distinction between Eq. 2 and 3 is indeed subtle and poorly thought out. On line 134, we specify that the expectation in Eq. 3 is wrt. a fixed distribution $p^\pi(\Lambda_{<i})$, from which $z_{i-1}$ and $\tau_{i-1}$ are drawn. This was meant to convey that rewards from option periods >= i had no bearing on stochastic decisions from option periods < i. In hindsight, this was a poor choice of notation. We will overhaul the notation in favor of a variant of Eq. 2 featuring discount factors, which are 1 during option execution, and 0 at option boundaries. This should make clear that no bootstrapping is performed across options.
> * L134 and L138: We will make explicit in Section 2, the joint distribution $p_\pi(\Gamma)$ and show how it decomposes as a product of conditional distributions for each option period (similar to how we define the joint $p^\pi(\tau)$ on line 50).
> * Using our notation, the hindsight trick can be expressed generally as \mathbb{E}_{p(x, y)} \mathbb{E}_{g(x \mid y)} [ f(x, y) ], for some approximate posterior $g$. We believe this is valid, but albeit loose notation. It should be interpreted as $\mathbb{E}_{x_1, y_1 \sim p(X, Y)} \mathbb{E}_{x_2 \sim g(X \mid Y=y_1)} [ f(x_2, y_1) ]$.
> * Usefulness is rather subjective in our opinion and the existence of [19, 12, 26, 22] is evidence that the KL-regularized RL, divergence minimization or "RL As Inference" perspectives are useful tools for unifying and in our opinion deriving novel algorithms for reinforcement learning, leveraging the toolbox of generative models and variational inference. As evidence of this being subjective, t5N6 favors a pure mutual information-based derivation, while this reviewer would favor one based purely on variational inference. We believe all three approaches are valid and would lead to the same result. We will add a note in the manuscript to this effect.
> ## [minor]
> * L256. Thank you for bringing this to our attention! The caption of Figure 4a should have read "(top) EDDICT-Delta (bottom) EDDICT". We will further add a direct link to Figure 4a on line 256, as being the experimental validation of this claim.
> * Sec 2: We are happy to switch from the set notation to sequence notation, to remove any potential source of confusion.
> * Fig 1 caption: We will clarify that a diamond denotes a deterministic quantity. As per lines 142-143, when inferring $z_i$ we condition on the expected value of $q_w(z_{i-1} \mid s_i)$.
> * Fig 4b. Thank you for the suggestion which will help to improve clarity. We will revise the figure accordingly.
> “The paper mentions that the method falls short of methods e.g. [10,11], yet these methods are not compared to. The paper needs to include more context for this statement (numbers!) as well as justify why more detailed comparison is not needed, or why the proposed method is still significant.”
>
> As stated in the paper, we fundamentally believe that head-to-head comparison should not be the focus of this work, as all of these methods learn a single exploratory policy, which have different potential applications compared to the conditional policies learned by EDDICT and other intrinsic control methods (e.g. DIAYN, VIC). For example, with a single policy it’s less clear how to do HRL, goal-achievement, and imitation; all of which were addressed in the DIAYN paper (amongst others e.g. RVIC) and can be broadly applied to this class of methods with minor modification. That said, in the updated paper we will include a table of previously published results so as to better contextualize EDDICT’s unsupervised performance on Atari.
>
>
> “The societal impact statement is embedded in the checklist -- instead, I think it should appear either in the main text or in the supplementary material.”
>
> We agree, and apologize for the oversight! Here is a draft of a societal impact statement to be included in the final version of the paper:
>
> “Unsupervised reinforcement learning in general, and intrinsic control methods in particular, are far from being commercialized due to their insufficient data efficiency and lack of validation in real world environments. However, when this is no longer the case, these methods could significantly reduce the human cost of setting up systems that interact with humans (e.g. robotics), as these methods limit the need for handcrafted reward functions and the collection of human preferences. But this benefit comes with a cost to interpretability and safety. The information theoretic objectives of the methods lead to behavior that can be very hard to predict a priori (e.g. what does ‘controlling your environment’ look like?). Furthermore, safety constraints might be harder to specify in the absence of a closed-form reward function. As these methods mature, the emphasis should shift from raw performance to a more nuanced approach that addresses these societal concerns head on.”

---

> > ### Comment · Reviewer_gr67 · 2021-08-28
> > **Response to response**
> >
> > Thank you for addressing my concerns. I'll increase my rating.

---

> > > ### Author Response · Authors · 2021-09-01
> > > **thank you**
> > >
> > > We really appreciate the score increase, and we will do everything in our power to make the final draft as improved as possible.

---

### Official Review · Reviewer_qvkL · 2021-07-16

**Rating:** 7
**Confidence:** 3

**Summary:**

This paper proposes an approach called Entropic Desired Dynamics for Intrinsic Control (EDDICT) which tries to learn options in an unsupervised manner. Given interaction with the environment, EDDICT aims to learn a latent code which best describes a future trajectory. This work is inspired from Variational Intrinsic Control (VIC). The authors argue that it is too difficult for a single latent code, $z$ to capture, so they propose a set of time-dependent latent codes. These time dependent codes are relative and follow a linear dynamics model. The authors show that the objective function for this setup (minimizing the KL divergence between the prior over trajectories, the distribution of latent codes conditioned on the trajectories and the posterior of observed trajectories) does not nicely break down into the same mutual information objectives as VIC but provide a greedy optimization based approach to solve this. They run toy examples on MuJoCo (pointmass) and on Atari games, and show strong exploratory performance.






**Limitations And Societal Impact:**

The authors do discuss limitations but do not discuss any negative societal impacts of their work. Self-supervised/unsupervised learning, especially when applied to robotics has many possible impacts (i.e. safety and interaction with humans, etc).

**Main Review:**

Strengths: This paper presents, to my knowledge, a novel idea of using time-variant latent codes for intrinsic control. The paper is grounded in theory and provides good explanations and intuition for why they pick this approach of optimizing for latent codes that change over time. EDDICT is well motivated as well, and in my opinion, approaches a problem of signficance. The idea of finding states that are "controllable" is definitely interesting and seems useful. The paper is also well written and the figures are informative. I find the pointmass toy experiment quite interesting, as EDDICT performs a lot better than VIC.


Weaknesses: While the paper presents a useful idea, I do have a few concerns. Firstly, it would be good for the authors to compare and contrast this approach with others that also try to find "controllable" states, such as Bisumulation based approaches [1]. Secondly, I question I have is that if the true dynamics model is highly non-linear, does the fact that latent codes follow a linear dynamics model hurt performance?

Additionally, the paper shows that EDDICT works well in point-mass regimes as well as in Atari, it would be good to see how this method performs on standard continuous control benchmarks such as DeepMind Control Suite and how well it performs against state-of-the-art unsupervised/self-supervised exploration methods such as [2], [3] etc.


[1] Learning Invariant Representations for Reinforcement Learning without Reconstruction. Zhang et al., 2020.
[2] Large-Scale Study of Curiosity-Driven Learning. Burda et al., 2019
[3] Go-Explore: a New Approach for Hard-Exploration Problems. Ecoffet et al., 2019.

**Time Spent Reviewing:**

5

---

> ### Author Response · Authors · 2021-08-11
> **Response to Reviewer qvkL**
>
> We thank the reviewer for their considerate and constructive feedback.
>
> “it would be good for the authors to compare and contrast this approach with others that also try to find "controllable" states, such as Bisumulation based approaches”
>
> We acknowledge that Bisimulation-based approaches have a similar motivation, but they differ in that they require an externally specified reward function in order to ground the notion of bisimulation, whereas our approach is fully unsupervised. That said, we will mention this in the discussion and cite the paper, as this distinction is non-obvious and worth mentioning.
>
> “Secondly, I question I have is that if the true dynamics model is highly non-linear, does the fact that latent codes follow a linear dynamics model hurt performance?”
>
> While this isn’t a very strong limitation in practice due to EDDICT’s ability to adapt the latent in which the dynamics are linear, it is certainly the case that many environments are not well modeled by EDDICT. Indeed, visualizing the latent space in Atari games shows that EDDICT’s representations are much less interpretable for games with sharp discontinuities (e.g. falling off a platform in Montezuma’s Revenge). We will include a discussion of this phenomena in the final draft, as well as present visualizations (similar to Figure 4a) showcasing failure modes. That said, even in environments that strain EDDICT’s representational capabilities, we still recover relevant structure (the representations are still interpretable in a local neighborhood) and obtain good exploratory behavior.
>
> “it would be good to see how this method performs on standard continuous control benchmarks such as DeepMind Control Suite”
>
> We agree! We chose Pointmass because there was an obvious “correct” representation (extract the x,y position of the agent from the image), but we have since run EDDICT on other DM Control tasks. We are at present somewhat limited by our base reinforcement algorithm (Q-learning with random candidate sampling for continuous actions) such that running on tasks with many action dimensions is problematic, but we will include visualizations akin to Figure 4a on several other tasks (e.g. half-cheetah and reacher). It is worth noting that EDDICT is technically agnostic to the underlying reinforcement learning algorithm, so tackling higher-dimensional action spaces found in the whole suite is a promising future direction.
>
> “how well it performs against state-of-the-art unsupervised/self-supervised exploration methods such
> as [ICM], [Go-Explore] etc.”
>
> Despite not yielding a family of conditional policies like our other baselines, ICM does learn an abstract transition model like EDDICT, so it is analogous in a sense. The followup to the initial ICM paper (“Large-Scale Study of Curiosity-Driven Learning”) also already includes performance on all of the Atari tasks we evaluate on, including our key result on Montezuma’s Revenge. While the methodology isn’t exactly the same (e.g. they only run a single seed for the unsupervised setting), we will include ICM for all Atari results with a footnote caveating the differences in methodology (Figure 5 and Table 1 in the main paper).
>
> Go-Explore is a bit of a corner case in that it exploits significant prior knowledge, namely full access to the simulator (e.g. jump to this state) and/or domain-specific state representations. While relaxing both of these constraints simultaneously might be feasible, this is not shown in either paper. Additionally modifying the algorithm for the unsupervised setting means that this represents a substantial contribution in itself. As such, a direct comparison is not in the scope of this work, but will be mentioned in the discussion.
>
> “The authors do discuss limitations but do not discuss any negative societal impacts of their work. Self-supervised/unsupervised learning, especially when applied to robotics has many possible impacts (i.e. safety and interaction with humans, etc).”
>
> We agree, and apologize for the oversight! Here is a draft of a societal impact statement to be included in the final version of the paper:
>
> “Unsupervised reinforcement learning in general, and intrinsic control methods in particular, are far from being commercialized due to their insufficient data efficiency and lack of validation in real world environments. However, when this is no longer the case, these methods could significantly reduce the human cost of setting up systems that interact with humans (e.g. robotics), as these methods limit the need for handcrafted reward functions and the collection of human preferences. But this benefit comes with a cost to interpretability and safety. The information theoretic objectives of the methods lead to behavior that can be very hard to predict a priori (e.g. what does ‘controlling your environment’ look like?). Furthermore, safety constraints might be harder to specify in the absence of a closed-form reward function. As these methods mature, the emphasis should shift from raw performance to a more nuanced approach that addresses these societal concerns head on.”

---

> > ### Comment · Reviewer_qvkL · 2021-09-01
> > **Post rebuttal comments**
> >
> > Dear authors,
> >
> > Thanks for your detailed response to my comments. I believe they have sufficiently addressed my issues, and I am increasting my score to a 7.
> >
> > Best regards

---

### Official Review · Reviewer_t5N6 · 2021-07-16

**Rating:** 6
**Confidence:** 4

**Summary:**

This work proposes a new method, Entropic Desired Dynamics for Intrinsic Control (EDDICT). EDDICT augments the Variational Intrinsic Control (VIC) algorithm with two main novelties: (1) replacing the distribution over latents/codes with a learned Markov chain $p(z_{i} | z_{i-1})$, and (2) using hindsight correction with the learned $q_z(|s)$ to stabilize the proposed objective. The Markov chain on codes enables latent codes to be "globally aware" by creating correlation between timesteps. Experiments show EDDICT can achieve better episodic state coverage, lifetime state coverage, and episodic return (of the unsupervised policy) compared to VIC and Relative VIC, as well as an ablated version where (1) is not included. The authors also empirically show the code norm can be inversely proportional to the state frequency, and that the the latent space structure on pointmass captures positional structure.

**Limitations And Societal Impact:**

Yes, the authors have adequately addressed the limitations of their work

**Main Review:**

Pros:
1. Authors propose an interesting new method which achieves better performance on the hard-exploration game Montezeuma's Revenge, particularly over the baseline VIC
2. Improvements in long-horizon application of skills is an important question in the research community, promising ability to perform longer horizon tasks
3. Experiments look technically sound
4. Insightful analyses in Figures 2/3 and 4

Cons:
1. I think the writing/notation was not very clear which led to confusion on my end; I may be wrong about some things but would appreciate if they could be clarified during the rebuttal period or in the text.
2. I think additional baselines, notably DIAYN [1], would help to strengthen the paper and the motivation
3. The mathematical derivation seems roundabout, in that the authors motivate the method as optimizing the KL divergence to a uniform distribution
4. Due to inconsistencies in the text (notably about how $q_w$ is defined and the AR(1) process, see below), and the lack of provided code, I think the paper does not contain enough details to be reproducible.

Overall, I think the paper explores an interesting idea that is interesting to consider. At this time, I would lean towards rejection due to clarity of the paper; however, if these are just misunderstandings on my end, I am open to changing my score:
1. A main question of mine is the formulation of the objective, i.e. Section 3.1 and Section 3.2 "EDDICT Posterior". Here, the objective is formulated as $-KL[p^\pi(\Lambda|s_0) || p^{\pi_{\emptyset}}(\Lambda | s_0)]$ (the equation above Equation 2). However, given that $\pi_{\emptyset}$ is taken to be uniform, I am not sure why this is interesting, or how it differs from the original VIC objective. I think it makes more sense to start from the VIC objective, $I(z; \tau | s_0)$ with the modification that now $p(z) = p(z_0) \prod_i p(z_i | z_{i-1})$, which would yield the same result after some similar assumptions.
2. In the same equation as above (the equation above Equation 2), there should be a $\geq$ instead of $=$ as introducing $q_w$ in place of $p$ makes the equation a variational lower bound.
3. In multiple places $q_w$ is defined differently: as $q_w(z|\tau)$ on line 79, as $q_w(z|s_0,s_T)$ on line 88, and as $q_w(z_{i-1}|s_i)$ on line 139 and Algorithm 1; I would appreciate if this were made consistent, and possibly would suggest using $q(z_i|\tau_i)$ if generality is desired before introducing the practical model $q_w(z_{i-1} | s_i)$ (if I am understanding them both correctly).
4. I would recommend replacing $\pi$ with $\pi_\theta$ when applicable, as otherwise for instance the RHS of Equation 2 seems to have no dependence on $\theta$.
5. Why does the intrinsic reward in Algorithm 1 drop the $-\log p(z_i | z_{i-1})$ term? This seems inconsistent with the text.
6. In many places in the text, AR(1) (citing Tirumala et al 2020) is cited. Given this is not a well-known idea and its definition is very short, I think it should be explained in the methods/preliminaries. A second thing is that in Algorithm 1, it appears AR(1) is not actually being used, and instead it is the "Learned AR process" from Tirumala et al. Also, $\mu_w(s_0)$ is undefined (and as written seems to share parameters with $q_w$).
7. Hindsight correction is an idea that is introduced, but I am not sure how it is shown in Algorithm 1.

Otherwise, I think the method is interesting, as inducing a prior/some correlation between latent codes makes sense. I do think, however, that this warrants a comparison to the style of DIAYN where a latent code is sampled at the beginning of the episode and not resampled. This would naturally enable a correlated skill for the entire episode, which EDDICT tries to do as well. I think it makes sense a well-tuned EDDICT could outperform this, as it resembles a middle ground between VIC and DIAYN, and that this experiment could strengthen the paper, but I don't think it can be taken for a given without experimental results. Furthermore, from the DIAYN paper, one strength of this approach is that the "discriminator looks at every state, which provides additional reward signal", which is lost in the EDDICT approach as $q_w$ only looks at the final state of each segment. I think this should be fairly easy to run simply by setting $K=1$ and removing the line resampling $z$.

Furthermore, I think the experiments/analysis from Figures 2/3 and 4 are quite interesting. However, I am not sure what the "target position" for the Pointmass experiment is (line 252) -- it seems to me we are not analyzing goal-conditioned algorithms. Figure 5/Table 1 is also promising, although I think it would be strengthened by the above proposed comparison to DIAYN, and possibly through contextualization to well-known exploration baselines such as RND [3].

Miscellaneous (no bearing on score)
1. Typo in "EDDICT Prior" section: $i \in [0, M[$
2. Figure 2/3 has some numbering issue
3. For Figure 2/3, I think it would make more clear to provide multiple examples of states in a given code frame, rather than taking the maximum over pixels like this.
4. I would appreciate if Figure 4 (a) had a legend (does yellow refer to the top of the environment?) -- it seems nontrivial a 2D position is being labeled with a 1D color

=====

References

[1] Eysenbach, Benjamin et al 2018. "Diversity is All You Need: Learning Skills without a Reward Function."

[2] Tirumala, Dhruva et al 2020. "Exploiting Hierarchy for Learning and Transfer in KL-regularized RL."

[3] Burda, Yuri et al 2018. "Exploration by Random Network Distillation."

=====

Updates

I have increased my score from 4 to 6, mainly due to the new comparison to DIAYN as well as other described improvements in clarity.

**Time Spent Reviewing:**

4

---

> ### Author Response · Authors · 2021-08-11
> **Response to Reviewer t5N6**
>
> We thank the reviewer for their detailed review and the positive feedback regarding the method, its analysis and recognizing the importance of learning useful skills over long horizons.
>
> We also  appreciate your concerns with respect to clarity of the paper and will strive to improve this for subsequent revisions. In the meantime, we trust the following will help address your concerns.
>
> [Cons2] We agree, and are now currently running 3 new DIAYN baselines. The first is DIAYN with the same code resampling setup we use for EDDICT and VIC: a new code is sampled at short fixed intervals without resetting the environment. The second new baseline is DIAYN with the code duration swept to over two orders of magnitude (20, 200, 2000), which should account for the fact that the DIAYN paper used longer code duration (though in a different environment). Finally, we are running versions of DIAYN and EDDICT with policy entropy terms (by switching the base RL algorithm from Q(\lambda) to soft-Q).
>
> The initial results of the first baseline, shown after 500M steps in the table linked [here](https://imgur.com/a/zMo9xEs), support our claim that EDDICT is not superseded by DIAYN, but might have complementary strengths. As can be seen in the table below, DIAYN has strong performance on a few of the games, but fails to match the progress on Montezuma’s Revenge, which remains our key result due to being well-known to provide a hard exploration challenge. We still need to sweep over entropy weight and skill duration, and will update this post as more results become available; we are particularly curious about the effect of policy entropy on both EDDICT and DIAYN.
>
> [Cons3, 1.] The KL-minimization perspective of VIC has already been established in [22 App B,C, 19]. While we could certainly derive EDDICT from a pure mutual-information based approach, we find the KL perspective more constructive, as it opens up the entire toolbox of graphical models. While it does seem odd at first glance to minimize this KL-divergence when $\pi_{\texttt 0}$ is uniform, the reverse predictor is itself part of the prior and is learnt as part of the same minimization process. The goal of minimizing this KL-divergence is thus not to learn a policy which generates trajectories having high-likelihood under a uniform policy – which would indeed be uninteresting by itself, but ones which further have high-probability under the reverse predictor, i.e. which are recognizable in hindsight. We will add this clarification to S3[.1] to further motivate our derivation. As for Eq 2 in particular, it differs from VIC in that it incorporates (i) option resampling vs the fixed episodic latent structure of VIC, and (ii) a conditional prior over $z_i$ in the form of a Markov chain.
>
> [Cons4] We hope this rebuttal will clear up some confusion and by extension improve reproducibility. While open-sourcing of the full experimental pipeline would be difficult due to closed-source dependencies, we are happy to provide an accompanying notebook, which would provide a canonical implementation of EDDICT, allowing the community to replicate the qualitative point mass experiment. We commit to open-sourcing this notebook by the camera-ready deadline.
>
> [2.] Thank you! You are correct that this should be an inequality. We will correct the manuscript accordingly.
>
> [3.] We realized our notation for the reverse predictor is a source of confusion. $q_w(z \mid \tau)$ should be interpreted as the general case, with $q_w(z \mid s_0, s_T)$ being the specific form employed by VIC, and $q_w(z_{i-1} | si)$ that employed by EDDICT. We propose using $q_w(z \mid \tau)$ (no resampling) or $q_w(z_i \mid \tau_i)$ (with resampling) for generality, and $q_w^{VIC}$ and $q_w^{EDDICT}$ when discussing the specific reverse predictor used by each algorithm.
>
> [4] We can see how this could be jarring, but thought $\pi_\theta$ and $\pi_{\texttt 0}$ would be prone to confusion. We will take this on board and change symbols for the prior policy to avoid confusion.
>
> [5] $z_i := z_{i-1} + \delta$, where \delta is drawn from a uniform distribution on the disk, hence $p(z_i|z_{i-1})$ is a constant and therefore unnecessary for the reward (adding a constant reward does not change the optimal policy or value functions). It would however be necessary for other distributions over $\delta$ (e.g. Gaussian, parametric distributions), and so adding it to Algorithm 1 would certainly make the algorithm more generic and future proof. We will make this change and discuss this in Section 3.4.
>
> [6] There is some confusion on whether the reviewer meant "well known" or "not well known". AR(1) is indeed a well established random process, and given that we use its simplest, parameter-less form of $X_t = X_{t-1} + \epsilon_t$, we may as well make this explicit from the start. Tirumala et al 2020 was cited since they also explore hierarchical policies whose prior over the codes is an AR(1) process. We can further include a textbook citation for the family of autoregressive random processes. $\mu_w(s)$ is defined in the comment to line 2, as the mean of the variational posterior $q_w(Z \mid s)$. Concretely, our reverse predictor is an isotropic Gaussian with mean $\mu_w(s)$. This additional symbol does seem superfluous, and we propose to use $z \leftarrow \Delta_z + \mathbb{E}[q_w(Z \mid s_0)]$ on line 2, and forego the comment entirely.
>
>  [7] Hindsight correction occurs on line 2, when we add $\Delta_z$ to the mean of our reverse predictor $\mu_w(s_0)$. We will add a comment to this effect in the algorithmic box. A possible source of confusion here is the change in notation from the single-stream non-episodic notation of $q_w(z_{i-1} | s_i)$ used throughout the main text – where $i$ indexes into the option period, and the algorithmic box which treats each option period as a (pseudo) episode, with initial state $s_0$ instead of $s_i$. Given this, line 2 should be interpreted as $z_i \leftarrow \Delta_z + z_{i-1}$, where we have substituted $z_{i-1}$ with $\mathbb{E}[q_w(Z \mid s_i)]$ (hindsight correction). We will change Algorithm 1 to use the same (single-stream, non-episodic) notation used throughout the rest of the paper, and apologize for the confusion.
>
> [target position, line 252] This refers to the Pointmass environment, as opposed to the solution method. The built-in reward function (which we don't use) for Pointmass involves moving the agent to a red dot on the screen that is randomly positioned between episodes. Hence, we refer to this red dot as the "target position", despite the fact it doesn't actually correspond to a goal in our experiments. Line 252 refers to the fact that EDDICT's representation doesn't capture this aspect of the environment, since it is outside of the agent's control. We claimed to be agnostic to the position of this distractor, but this is not obviously the case from the present figure. As such, we’re adding a new figure to the appendix depicting the effect of varying the distractor position on the latent representation while holding the point-mass’ position constant (there is no effect).

---

> > ### Comment · Reviewer_t5N6 · 2021-08-22
> > **Response to rebuttal**
> >
> > Thank you for the detailed rebuttal! I have increased by score from 4 to 6. I think the added comparison to DIAYN greatly strengthens the experimental section, as DIAYN represents a strong modern baseline. Combined with the existing experimental section, improvements to clarity, and the responses to the other reviewers, I now believe this paper meets the bar for publication.

---

> > > ### Author Response · Authors · 2021-09-01
> > > **Thank you**
> > >
> > > Thank you so much for the score increase. We really appreciate it, and promise we use everything we learned during this discussion to improve the clarity of the final draft.

---

### Official Review · Reviewer_nvNV · 2021-07-18

**Rating:** 6
**Confidence:** 3

**Summary:**

This paper aims to solve the exploration issue in existing mutual information based intrinsic control methods.
The issue is insufficient exploration because using a ﬁxed prior of latent codes means that such approaches are unable to learn codes that correspond to states that cannot be reached in the time horizon T, since any code can be sampled in any state.

The authors propose to replace the fixed prior distribution of z with a fixed dynamics model over z.

To make the objective tractable, the authors start from the divergence minimization perspective of VIC and derive a lower bound posterior. They further incorporate two design choices with the optimization of the posterior to ensure learning diverse linear dynamic codes.

**Limitations And Societal Impact:**

Yes

**Main Review:**

While the studied problem is important and the propose method is sensible, experiments are insufficient and not solid enough.

ICM [Pathak et al] is a very related baseline since the representation learned there is also about controllable features and similarly it uses this controllable representation for exploration.

For the coverage/exploration evaluation, the authors should compare it against CPT [Campos et al], APT [Liu et al] and RND [Burda et al].

It should also be made clear about the semantic meaning of “average episodic coverage” in Figure 5, e.g., how significantly it is to achieve 6000? One possible way to do this is to show the results from exploration methods, as mentioned above.

**Time Spent Reviewing:**

3

---

> ### Author Response · Authors · 2021-08-11
> **Response to Reviewer nvNV**
>
> We thank the reviewer for their constructive feedback and suggestions.
>
> “For the coverage/exploration evaluation, the authors should compare it against CPT [Campos et al], APT [Liu et al] and RND [Burda et al].”
>
> We wanted our primary comparisons to be against methods that were also learning conditional policies, as these methods differ in their potential downstream usages compared to methods like ICM, APT, and RND that learn a single policy (e.g. HRL, goal achievement). We will however, add a figure to the appendix that lists the previously published unsupervised reward achievement on Montezuma’s Revenge, including all of these methods. We aren’t looking to claim state of the art on this particular metric, but we hope this additional information will help contextualize our results.
>
> “ICM [Pathak et al] is a very related baseline since the representation learned there is also about controllable features and similarly it uses this controllable representation for exploration.”
>
> That said, ICM does learn an abstract transition model like EDDICT, so it is analogous in other ways. The followup to the initial ICM paper (“Large-Scale Study of Curiosity-Driven Learning”) also already includes performance on all of the Atari tasks we evaluate on, including our key experiment on Montezuma’s Revenge. While the methodology isn’t exactly the same (e.g. they only run a single seed for the unsupervised setting), we will include ICM for all Atari results with a footnote caveating the differences in methodology (Figure 5 and Table 1 in the main paper).
>
> “It should also be made clear about the semantic meaning of “average episodic coverage” in Figure 5, e.g., how significantly it is to achieve 6000?”
>
> Our coverage metric translates to the number of unique avatar positions. We’ve added a [DIAYN baseline](https://imgur.com/a/zMo9xEs), which should hopefully add some context as to how other methods fare. And we are also a random agent baseline, which should serve as a lower-bound on agent coverage performance.

---

> > ### Comment · Reviewer_nvNV · 2021-08-25
> > **Reply to author response**
> >
> > Thank you for providing random agent baseline to help contextualize the episodic coverage results. I really hope the authors can make the promised experiments to the next updated version. In light of these changes, I have increased the score from 5 to 6.

---

> > > ### Author Response · Authors · 2021-09-01
> > > **Thank you**
> > >
> > > Thank you for increasing your score, we greatly appreciate it.
> > >
> > > [Here](https://imgur.com/a/o5ji58K) are the results for two random policy baselines. These involve running either a random deterministic policy (i.e. freeze network at init and run greedy policy) or stochastic (freeze network at initial and run softmax policy). These scores are generally quite low, but do give the relative impressiveness of the other results. We will be adding the table to the appendix, and are thinking planning to average the average of these random baselines to normalize the main results figure for the final draft.

---

### Official Review · Reviewer_GDC1 · 2021-07-21

**Rating:** 7
**Confidence:** 3

**Summary:**

This paper proposes an intrinsic exploration method, entropic desired dynamics for intrinsic control (EDDICT) to learn globally and temporally consistent latent exploration codes. Compared to variational intrinsic control (VIC), EDDICT proposes a number of changes driven by additionally reasoning over sub-trajectories and adding p(z_i+1|z_i) as a factor to the graphical model. Adding this factor makes the latent code local and therefore potentially easier to learn a policy for, but now the algorithm must reason over multiple latent codes per full trajectory. EDDICT proposes to optimize it greedily (only considering the entropy using the next latent code) but ties the latent structure together globally by choosing linear AR(1) dynamics for z_i and conditioning on the most likely past z (hindsight correction). The experimental results show that EDDICT outperforms prior exploration methods across 6 difficult Atari games including successfully solving Montezuma’s revenge.

**Limitations And Societal Impact:**

Limitations are discussed in the discussion section.

**Main Review:**

EDDICT is motivated from the viewpoint of learning globally and temporally consistent latent exploration codes. Starting with the divergence-minimization view of VIC, the method is derived by adding a temporal factor p(z_i|z_{i-1}). To deal with the harder optimization with the extra factor, EDDICT proposes three changes: greedy optimization w.r.t. only one option, hindsight correction to use the most likely previous latent, and simple linear dynamics of the latent code. The paper also provides a kind of template for designing better exploration methods by adding constraints and assumptions.

The experiments show that EDDICT successfully learns globally coherent latent codes, and explores without rewards successfully in difficult Atari environments including Montezuma’s revenge.

For globally coherent codes, EDDICT and VIC are evaluated on a pointmass environment with a U-shaped wall. EDDICT with a 2D latent code learns a globally coherent mapping of the states in the environment which corresponds directly to x and y position of the pointmass with a hole in representation due to the wall. In comparison, VIC learns a much more entangled embedding.

Then, EDDICT is shown to successfully solve hard exploration Atari games, with an emphasis on Montezuma’s revenge. The results here seem very impressive: EDDICT is the only method that really makes significant progress on Montezuma’s revenge in terms of average return from the ground truth reward (that methods do not have access to), as well as coverage metric (note, see Figure 1 in the appendix *not* the main paper). Interestingly, the only other method that makes some nontrivial progress and is also somewhat competitive in the other 5 Atari games is RVIC. These experiments additionally include EDDICT-delta which if I understood correctly has independent latent codes instead of a temporally connected AR(1) process. EDDICT-delta performs poorly; I inferred this means that connecting z temporally is important, but the conclusion does not seem explained in the text. It would have also been good to include ablations of the greedy optimization and hindsight correction to better understand each contribution individually.

The coverage of prior seems relatively complete. One thing I would have like to see is comparison to other exploration methods outside latent code methods - for instance, one strong candidate is Go-Explore [1], which also achieves strong performance on Montezuma’s revenge, and also a comparison to curiosity/novelty-based methods which are discussed but not compared.

[1] First return, then explore. Ecoffet 2020.

Minor

Line 150: p(z_i|z_{i+1}) = z_{i-1} + \delta_i should be z_i = … instead of p( ) = …

Table 1 is a bit hard to parse, maybe turn into something visual?

**Time Spent Reviewing:**

6

---

> ### Author Response · Authors · 2021-08-11
> **Response to Reviewer GDC1**
>
> We thank the reviewer for their insightful comments and constructive suggestions.
>
> “Figure 1 in the appendix not the main paper”
> We apologize for this issue -- we caught a bug in our coverage metrics and were unable to include this experiment by the main paper deadline, only the appendix deadline (which was a week later). We assure reviewers that Figure 5 in the main paper has been updated to reflect the bug-free experiment shown in Figure 1 in the appendix.
>
> “These experiments additionally include EDDICT-delta which if I understood correctly has independent latent codes instead of a temporally connected AR(1) process. EDDICT-delta performs poorly; I inferred this means that connecting z temporally is important, but the conclusion does not seem explained in the text.”
>
> Indeed, this was the intent of the EDDICT-delta ablation, and we agree that this should be stated more explicitly in the text -- we are rewriting the affected sections to address this. It is also worth noting that due to a typo, Figure 4a claims we’re comparing EDDICT to VIC on Pointmass, when we were actually comparing to EDDICT-delta. These visualizations provide additional qualitative evidence that connecting z temporally is desirable.
>
> “include ablations of the greedy optimization and hindsight correction to better understand each contribution individually.”
>
> We actually already ran these, but didn’t initially include them because of their extremely poor performance. Without hindsight correction, z is updated in an open-loop manner and quickly diverges from the corresponding observations. The ablation without greedy optimization fails for more complex reasons, likely due to bootstrapping over multiple non-stationary rewards and the increased effective time horizon. While the reasoning for this failure is speculative, we note that this failure was also observed in VIC and RVIC. Indeed, the RVIC paper acknowledges the usage of greedy optimization, though they state this in more concrete terms: “0 γ at skill end” for Atari (in Appendix A). As such, we posit that understanding the need for greedy optimization is not unique to EDDICT, and remains an open question for all intrinsic control methods. We will update the discussion section to reflect this.
>
>
> We agree that despite their uninteresting empirical performance (close to floor for all experiments), these ablations are worth including as they ground our motivations in concrete empirical results. As such, we will add them to Figure 5 in the main text (Montezuma Revenge results) and Table 2 in the appendix (6 game results).
>
> “One thing I would have like to see is comparison to other exploration methods outside latent code methods - for instance, one strong candidate is Go-Explore [1], which also achieves strong performance on Montezuma’s revenge, and also a comparison to curiosity/novelty-based methods which are discussed but not compared.”
>
> We agree that the relative merits of curiosity/novelty-based methods is of interest, but intrinsic control / latent code methods have a different (though certainly overlapping) set of use cases from novelty-based approaches, due to learning a family of policies (i.e. code conditional) instead of a single policy. As mentioned in the discussion, learning a single policy appears to currently be more scalable in terms of just producing exploratory behavior, but to make this concrete we’ve attached a new table to the appendix that lists published unsupervised reward attainment on Montezuma’s Revenge for several novelty-based methods (NGU, RND, ICM).
>
> Go-Explore is a bit of a corner case in that it exploits significant prior knowledge, namely full access to the simulator (e.g. jump to this state) and/or domain-specific state representations. While relaxing both of these constraints simultaneously might be feasible, this is not shown in either paper. Additionally modifying the algorithm for the unsupervised setting means that this represents a substantial contribution in itself. As such, a direct comparison is not in the scope of this work, but will be mentioned in the discussion.
>
> In terms of specific use-cases, the simplest case for EDDICT would be to learn it alongside a task reward and use the skills for exploration, and we’ve run a proof-of-concept of this in Montezuma. It can consistently reach a score of 2.4k in under 100M frames, which is competitive with existing SOTA novelty and option-learning approaches in this data regime (Machado et al, 2020).
>
> Machado, M. C., Bellemare, M. G., & Bowling, M. (2020, April). Count-based exploration with the successor representation. In Proceedings of the AAAI Conference on Artificial Intelligence (Vol. 34, No. 04, pp. 5125-5133).

---

### Author Response · Authors · 2021-08-10
**General Rebuttal (complementary to forthcoming reviewer-specific responses)**

We’ve collected all of the things we’ve promised reviewers below. We commit to adding the results of 1-5 as they come in during the discussion period (via anonymous imgur links), and we commit to carrying out the full set of changes in the final version of the paper, with open-sourcing to be completed prior to the conference date.

1) [DIAYN baseline](https://imgur.com/a/zMo9xEs), sweeping entropy weight and goal duration
2) EDDICT with dense rewards and policy entropy (akin to DIAYN)
3) [Random policy baseline to contextualize the coverage metric](https://imgur.com/a/o5ji58K)
4) EDDICT ablations for hindsight correction and greedy optimization
5) Co-training EDDICT and task reward on Montezuma’s Revenge ([learning curve](https://imgur.com/a/0Dspd2a))
6) Open-source a stand-alone version of EDDICT for running on toy environments
7) Qualitative results other DM Control Suite tasks (HalfCheetah, Reacher)
8) New figure showing EDDICT's representations don't capture uncontrolled distractor in PointMass
9) Additional version of Figure 3(right), where images are replaced by 2D histogram of avatar screen position
10) New table comparing previously published unsupervised scores on Montezuma’s Revenge
11) Comparison to ICM in the main results figures
12) Add societal implications section

# Comparison to single policy exploration methods.

Several reviewers brought up the relationship between EDDICT and other purely exploration focused methods, such as RND [nvNV t5N6], CPT [nvNV] and intrinsic curiosity methods [GDC1,nvNV, qvkL], e.g. ICM. As stated in the paper, we fundamentally believe that head-to-head comparison should not be the focus of this work, as all of these methods learn a single exploratory policy, which have different potential applications compared to the conditional policies learned by EDDICT and other intrinsic control methods (e.g. DIAYN, VIC). For example, with a single policy it’s less clear how to do HRL, goal-achievement, and imitation; all of which were addressed in the DIAYN paper (amongst others e.g. RVIC) and can be broadly applied to this class of methods with minor modification. That said, in the updated paper we will include a table of previously published results so as to better contextualize EDDICT’s unsupervised performance on Atari.


# Comparison to DIAYN

Diversity Is All You Need is a prominent intrinsic control method, which was not initially used as a baseline in this paper. Our position was that it was sufficiently similar to Variational Intrinsic Control (VIC) so as to obviate its inclusion. But several reviewers point out that its dense reward signal might significantly change performance relative to our other baselines. We agree, and are now currently running 3 new baselines. The first is DIAYN with the same code resampling setup we use for EDDICT and VIC: a new code is sampled at short fixed intervals without resetting the environment. The second new baseline is DIAYN with the code duration swept to over two orders of magnitude (20, 200, 2000), which should account for the fact that the DIAYN paper used longer code duration (though in a different environment). Finally, we are running versions of DIAYN and EDDICT with policy entropy terms (by switching the base RL algorithm from Q(\lambda) to soft-Q).

The initial results of the first baseline, shown after 500M steps in the table ([linked here](https://imgur.com/a/zMo9xEs)), support our claim that EDDICT is not superseded by DIAYN, but might have complementary strengths. As can be seen in the table below, DIAYN has strong performance on a few of the games, but fails to match the progress on Montezuma’s Revenge, which remains our key result due to being well-known to provide a hard exploration challenge. We still need to sweep over entropy weight and skill duration, and will update this post as more results become available; we are particularly curious about the effect of policy entropy on both EDDICT and DIAYN.

# Utility in conjunction with extrinsic rewards

We strongly believe that learning a rich set of skills which are tailored to the particular environment dynamics, but agnostic to the reward function is key to improving data efficiency in reinforcement learning. This paper is primarily concerned with the first step in this direction, improving on SoTA intrinsic control methods by introducing temporally correlated skills which improve coverage and improved representations. While we do believe the resulting options could be used as part of a full HRL agent (as done in DIAYN), we believe this is best left as a follow-up paper. We do have reason to be optimistic however: by co-training an exploration policy with EDDICT, a standard RL agent maximizing extrinsic rewards can achieve up to 2.5k results on Montezuma in around 100M steps ([learning curve shown here](https://imgur.com/a/0Dspd2a)).

# Code release

Reviewer t5N6 raised the point that the replicability of EDDICT would benefit from a concrete implementation. We agree wholeheartedly, and commit to releasing a stand-alone IPython notebook, prior to the conference in December, that will allow anyone to run simple experiments that replicate our claims about EDDICT’s representations (e.g. distractor robustness), exploratory behavior and utility in conjunction with task rewards.

# Effect of Distractors

Reviewers t5N6 and gr67 both expressed concerns about the qualitative Point Mass experiments which we believe to be of general interest. There was a semantic typo in Figure 4a (caught by gr67), wherein we wrote we’re comparing to VIC, when we were actually comparing to the ablated version of EDDICT (EDDICT-\delta). With this correction in place, we hope that the intent and findings are clear, i.e.: this experiment demonstrates the necessity of the latent dynamics (as this was all that was removed by the ablation). The other claim in this experiment is the demonstration that EDDICT inherits the property (shared by all intrinsic control methods) of being agnostic to uncontrollable aspects of the environment. The Point Mass environment naturally has one such distractor in the form of the “target position” -- a static red circle whose location is randomly determined every 1000 steps. We claimed to be agnostic to the position of this distractor, but this is not obviously the case from the present Figure. As such, we’re adding a new figure to the appendix depicting the effect of varying the distractor position on the latent representation while holding the point-mass’ position constant (there is no effect).

# Limitations And Societal Impact

Reviewers gr67 and qvkL mention that our discussion of societal impact is limited to the checklist, and that some further discussion is warranted. We concur, and here is the additional statement we will be including in the final version of the paper:

Unsupervised reinforcement learning in general, and intrinsic control methods in particular, are far from being commercialized due to their insufficient data efficiency and lack of validation in real world environments. However, when this is no longer the case, these methods could significantly reduce the human cost of setting up systems that interact with humans (e.g. robotics), as these methods limit the need for handcrafted reward functions and the collection of human preferences. But this benefit comes with a cost to interpretability and safety. The information theoretic objectives of the methods lead to behavior that can be very hard to predict a priori (e.g. what does ‘controlling your environment’ look like?). Furthermore, safety constraints might be harder to specify in the absence of a closed-form reward function. As these methods mature, the emphasis should shift from raw performance to a more nuanced approach that addresses these societal concerns head on.

---

> ### Author Response · Authors · 2021-09-01
> **Additional seeds**
>
> While 3 seeds is on par with other work in this field, during this discussion period we have been able to run more.
>
> We have run additional experiments to bring the total seeds per condition to 10. Here is a [link](https://imgur.com/a/DPr5Xrv) to the new results, with only clear winners bolded (i.e. no overlap in the confidence intervals). While there are still fewer clear winners than under the more liberal bolding scheme (i.e. bold the highest), EDDICT is now the only method significantly beats all other baselines on any game by any metric. Indeed, EDDICT wins by at least one metric on 4 of the 6 games.
>
> We also commit to running 10 seeds for VIC and best DIAYN variant (updating the table for the final draft), but were unable to do so during the response period due to resource limitations. We prioritised RVIC and the EDDICT ablation because our initial results suggested that they were the strongest baselines.
>
> We hope this added rigor bolsters confidence in our results, and we thank the reviewers for their lively discussion.

---

### Decision · Program_Chairs · 2021-09-28

**Decision:**

Accept (Poster)

**Comment:**

This paper studies the problem of making exploration efficient by learning a global-local structure. The paper generally received positive reviews which tended towards acceptance. However, the reviewers had difficulty understanding some notations and suggested comparing the proposed method to some relevant baselines. The authors provided a rebuttal that addressed many of the reviewers' concerns. The paper was discussed post rebuttal and all the reviewers responded to the rebuttal. Reviewers generally agree that the paper should be accepted but there are still many pending updates that authors have promised to finish. AC agrees with the reviewers and suggests acceptance. The authors are urged to look at reviewers' final feedback and incorporate the new experiments/baselines in the camera-ready.

**Consistency Experiment:**

NeurIPS has a long history of experimentation. In 2014, NeurIPS ran an experiment in which 10% of submissions were reviewed by two independent committees to quantify the randomness in the review process. This year, we repeated a variant of this experiment to see how the quality of the review process has changed over time.  This paper was part of the experiment and was therefore assigned to two committees (consisting of reviewers, an Area Chair, and a Senior Area Chair) that reached independent decisions.  If both committees made the same recommendation, this recommendation was followed. If a single committee recommended acceptance, the paper was accepted (with the exception of a few cases in which the other committee identified what we considered a fatal flaw, e.g., an error in a key result).

This copy’s committee reached the following decision: **Accept (Poster)**

The other committee assigned to the paper recommended **Reject**.  You can find the other set of reviews, along with any follow up discussion with the authors here:
https://openreview.net/forum?id=lBSSxTgXmiK